# Solar radiative transfer in Antarctic blue ice: spectral considerations, subsurface enhancement, inclusions and meteorites

Andrew R.D. Smedley[1,2], Geoffrey W. Evatt[2], Amy Mallinson[2], and Eleanor Harvey[2,3]

[1] Department of Earth and Environmental Sciences, University of Manchester, Manchester, M13 9PL, UK
[2] Department of Mathematics, University of Manchester, Manchester, M13 9PL, UK
[3] EPSRC Centre for Doctoral Training in Fluid Dynamics, University of Leeds, Leeds, LS2 9JT, UK

*Correspondence to*: Andrew R.D. Smedley (andrew.smedley@manchester.ac.uk)

**Abstract.** We describe and validate a Monte Carlo model to track photons over the full range of solar wavelengths as they travel into optically thick Antarctic blue ice. The model considers both reflection and transmission of radiation at the surface
of blue ice, scattering by air bubbles within it and spectral absorption due to the ice. The ice surface is treated as planar whilst bubbles are considered as spherical scattering centres using the Henyey-Greenstein approximation. Using bubble radii and number concentrations that are representative of Antarctic blue ice, we calculate spectral albedos and spectrally-integrated downwelling and upwelling radiative fluxes as functions of depth and find that, relative to the incident irradiance, there is a marked subsurface enhancement in the downwelling flux and accordingly also in the mean irradiance. This is due
to the interaction between the refractive air-ice interface and the scattering interior and is particularly notable at blue and UV wavelengths which correspond to the minimum of the absorption spectrum of ice. In contrast the absorption path length at IR wavelengths is short and consequently the attenuation is more complex than can be described by a simple Lambert-Beer style exponential decay law — instead we present a triple exponential fit to the net irradiance against depth. We find that there is a moderate dependence on the solar zenith angle and surface conditions such as altitude and cloud optical depth.
Representative broadband albedos for blue ice are calculated in the range 0.585 to 0.621. For macroscopic absorbing inclusions we observe both geometry- and size-dependent self-shadowing that reduces the fractional irradiance incident on an inclusion's surface. Despite this, the inclusions act as local photon sinks and are subject to fluxes that are several times the magnitude of the single scattering contribution. Such enhancement may have consequences for the energy budget in regions of the cryosphere where particulates are present near the surface. These results also have particular relevance to
measurements of the internal radiation field: account must be taken of both self-shadowing and the optical effect of introducing the detector. Turning to the particular example of englacial meteorites, our modelling predicts iron meteorites to reside at much reduced depths than previously suggested in the literature (< 10 cm *vs* ~40 cm), and further shows a size dependency that may explain the observed bias in their Antarctic size distribution.

# 1 Introduction

Incident solar radiation varies over a range of timescales due to the predictable seasonal and daily motion of the sun-earth system, supra-daily stochastic influences of the changing atmosphere and longer-term climate effects. It is a key driver of the cryosphere's energy budget (van den Broeke et al., 2011; Van Tricht et al., 2015; Hofer et al., 2017) and its variability

strongly effects the internal temperature profile of ice, particularly close to the surface (Liston et al., 1999). This has implications for processes such as ice sheet near surface melting and ice shelf crack formation (Bennartz et al., 2013; van den Broeke et al., 2016; Webster et al., 2017). In addition to driving physical processes, the quantity and spectral composition of solar radiation transmitted through ice, or available within it, can have significant effects on aquatic and other polar ecosystems.

The present study was specifically initiated in response to the need for a better understanding of this glacial subsurface radiative field. Recently Evatt et al. (2016) presented a mathematical model for the vertical movement of meteorites through blue ice in Antarctica. In this work the attenuation of solar radiation through ice and the absorption of solar radiation by a meteorite were modelled using the Lambert-Beer law. Although this approach works well at certain depths, it is limited in its accuracy, particularly near the surface. This issue specifically motivated us to find a more accurate, yet still simple and easily

applicable, methodology for modelling the attenuation and absorption of solar radiation through blue ice.

Despite this specific motivation, there are obvious parallels that can be drawn with several different climatologically important research foci. Examples include the impact of anthropogenic soot, pollutants, cryoconites and other englacial absorbers near the surface of the Greenland Ice Sheet (e.g. Box et al., 2012; Dumont et al., 2014; Stibal et al., 2017). In all of these cases, including that of englacial meteorites, the inclusions have a low albedo and are subject to an atmospherically-

modulated near-surface radiation field, cause local heating and thus can contribute to increased melt rates.

Notwithstanding the importance of shortwave radiative transfer in the aforementioned studies there is a tendency, as in Evatt et al. (2016), to treat the shortwave radiative flux as a single broadband parameter (via the Lambert-Beer law) and neglect to incorporate the range of behaviours exhibited by different wavelengths of solar radiation. This is a fundamental simplification as the incident solar spectrum exhibits a great deal of structure due to terrestrial and solar processes, whilst the

absorption spectrum of ice spans eight orders of magnitude across the solar wavelength range (Brandt and Warren, 1993; Warren and Brandt, 2008). For some applications such a simplification is reasonable, however for others — such as those concerned with the near-surface heat budget — a more encompassing model will be beneficial.

It is also instructive to briefly compare, in order of increasing density, the optical characteristics of Antarctic blue ice to other cryospheric forms of water. Freshly fallen snow is loosely packed and because it is formed of many irregular voids, often

characterised in terms of its grain size. As suggested by its high albedo, it is highly scattering. Thus even physically thin snowfalls usually are optically thick. As it graduates to firn, that is, partially recrystallised older snow that has generally survived a melt season, it becomes denser and contains fewer voids. Like snow, firn is usually optically thick, but has a longer scattering path. Sea and lake ice exhibit scattering centres that are comprised of bubbles, both often exhibit a layered

structure and are additionally underlain by low scattering parent bodies of water, though sea ice contains brine pockets affecting its optical properties. In contrast blue ice is characterised by its colour and is formed through long-term compression of snow. It has a high density, is physically and optically thick, and exhibits a longer scattering path than the other forms due to relatively few scattering centres (bubbles). Accordingly its bulk optical properties are the result of both

the intrinsic absorption by ice and the residual scattering by remnant bubbles.

There are three key studies that have investigated radiative transfer within various types of ice in detail and gave due attention to the aforementioned spectral issues. Mullen and Warren (1988) developed a radiative transfer model of lake ice to illustrate the processes responsible for the resulting albedo and transmission through a layer of ice. They treated bubbles as spheres, deriving the scattering coefficient and asymmetry parameter from Mie calculations, and relied on the delta-

Eddington method in their treatment of multiple scattering. They showed results across the solar waveband for the direct beam and diffuse incidence as well as the transmission for different bubble concentrations. Later, Light et al. (2003) developed a Monte Carlo model for radiative transfer in sea ice. Their focus was on cylindrical samples of ice, with the model being used to interpret backscattering from cylindrical core samples. In both these studies the emphasis is on relatively optically thin samples where the ice overlays a body of water, or where the parameter of interest is the

transmission. In cases where ice overlays water there is very little change in the refractive index at the lower ice-water interface: this effectively removes any lower refractive boundary and permits downwelling photons to continue their trajectory into the water with reduced opportunities for further scattering. Therefore an assessment of the transmission and reflectance of the incident sunlight is considered an adequate summary of the interaction in these cases.

The third key study was presented by Liston et al. (1999) and relates most closely to that described here as it detailed the ice

melt in blue ice *vs*. deep snow areas. The authors' focus, however, was on understanding the resulting subsurface temperature and melting profiles, and they relied on a two-stream approach constrained by specific atmospheric forcings and measured surface albedos. Account was taken of the spectral nature of the problem; however to maintain consistency between the treatment of snow and blue ice the optical properties were linked to effective grain size and a spectral extinction coefficient was calculated on this basis. We also note related studies that investigated the spectral albedo of white sea ice or

snow, but not the internal radiative field (Gardner and Sharp, 2010, Ehn et al., 2011; Haussener et al., 2012; Malinka et al., 2016; Taskjelle et al., 2017), or considered internal scattering from a purely theoretical perspective (Malinka, 2014). However to our knowledge there has been no study for optically thick blue ice where the spectral radiative transfer and albedos are derived from Monte Carlo modelling of solar radiation interacting with embedded bubbles and the underlying material properties. In the present study we therefore take this approach to address two core aims. The first is to present an

in-depth investigation of the radiation field within optically thick bubbled ice at different solar wavelengths, including a range of sensitivity tests and its impact on inclusions. This then leads us to the second aim: a distillation of these results into a simple, and widely applicable, mathematical model for the net flux.

In Sect. 2 we describe the details of our Monte Carlo radiative transfer model, including its initialisation, the conditions at the boundaries, the scattering of photons by bubbles and the eventual absorption of photons by ice. The validation of the

model is also discussed. In Appendices A and B we describe our methodology for the related calculations of the incident solar spectrum and the derivation of the relevant bubble parameters. In Sect. 3 the model is used to investigate how the incident solar spectrum propagates through ice, both when considered spectrally and when integrated across solar wavelengths. We investigate the influence of varying the bubble number concentration and effective radius, the effect of

varying the solar zenith angle and the influence of different surface environments which might alter the spectral balance of the incident solar spectrum. In Sect. 4 a macroscopic inclusion (absorbing target) is added to the model in order to study how the target's geometry and size impact on the effective radiation field incident on its surface. In Sect. 5, curves are fit to the integral shortwave results for the net radiative flux in a typical blue ice area. Whilst, lastly, in Sect. 6 we discuss how these results relate to the specific application of modelling the dynamics of meteorites in Antarctic blue ice.

**2 Monte Carlo model description**

Whilst different approaches to utilising the general radiative transfer equation exist (for background see Thomas and Stamnes, 1999), here we choose to use an unpolarised Monte Carlo simulation approach to investigate radiative transfer within bubbled ice. We do so for its ability to represent the different physical aspects, its more intuitive nature and its capability to study inclusions in a non-plane parallel fashion.

The core Monte Carlo model aims to calculate the downwelling and upwelling solar irradiance fluxes, $E_\downarrow$ and $E_\uparrow$, through bubbled ice as a function of depth by tracking the random walk pathways of simulated photons through the medium. Photons incident on the ice surface arrive from the atmosphere, and if not reflected by the interface, pass into the ice. Photons that enter the ice are either scattered by the air bubbles trapped within it, or are absorbed. Macroscopically the ice is considered as a semi-infinite slab with no lower boundary. Within this slab we follow the approach of Mullen and Warren (1988) and

model a distribution of bubbles of effective radius $r_{bub}$ and number concentration $N_{bub}$. Figure 1 shows a schematic of the model geometry. Our initial focus is on the downwelling and upwelling irradiance as they encapsulate more information about the internal radiation field than the net irradiance or the absorption profile with depth do, whilst still allowing these quantities to be derived as necessary. However, the net irradiance (flux) is clearly important in a glaciological framework and we will turn our attention to it in Sect. 5. We define the fractional irradiance as the fraction of photons that are in flight

(not yet absorbed) and hence contribute at a given depth below the surface, whether downwelling or upwelling. Both are normalised by the incident surface irradiance, $E_0$ and so are given by $E_\downarrow/E_0$ and $E_\uparrow/E_0$ respectively. Accordingly, absolute values of the upwelling and downwelling components can be calculated by multiplying the fractional quantities by the incident surface irradiance. Likewise the spectral albedo is defined at each wavelength, $\lambda$, as the fraction of incident photons that escape back to the atmosphere, either by direct reflection or internal scattering; this is calculated under a diffuse sky

except where noted otherwise.

## 2.1 Model inputs

There are three principal sets of inputs to the Monte Carlo model: the incident solar spectrum, the intrinsic optical properties of bubbled ice, and the geometric characteristics of the bubbles. These are detailed in turn below.

The incident spectral irradiance at the ice surface is calculated using the libRadtran radiative transfer model (Mayer and Kylling, 2005) with relevant atmospheric inputs for clouds, aerosols, and solar zenith angle. These, and the surface altitude and broadband albedo, are initially chosen to be appropriate for a blue ice area near the Frontier Mountain range, Antarctica [72.95° S, 160.48° E]. The inputs are similar to those Evatt et al. (2016) used to calculate a climatology of integrated shortwave fluxes, however for completeness they are described in some more detail in Appendix A.

The intrinsic optical properties of the ice are fully defined by the wavelength of the photon being tracked ($\lambda$), $r_{bub}$, and $N_{bub}$ as follows. The complex refractive index is taken from Warren and Brandt (2008) with the real part of the refractive index data, $m_{re}$, being interpolated at intermediate wavelengths ($\lambda$) linearly in $\log(\lambda)$, and the imaginary part, $m_{im}$, being interpolated to intermediate wavelengths on a log-log basis. The scattering and absorption coefficients, $k_{sca}$ and $k_{abs}$, are then expressed in terms of the complex refractive index, the bubble effective radius, and the number concentration, as follows (Mullen and Warren, 1988; Hecht, 1987):

$$k_{sca} = 2\pi r_{bub}^2 N_{bub}, \tag{1}$$

$$k_{abs} = \frac{4\pi m_{im}}{\lambda}\left(1 - \frac{4\pi r_{bub}^3 N_{bub}}{3}\right), \tag{2}$$

where the bracketed term in Eq. 2 accounts for the porosity of the ice.

Turning to the geometric characteristics of blue ice bubbles, there is a paucity of relevant in-situ data and thus we principally rely on a homogenisation of bubble density and radii concentration measurements carried out by Dadic et al. (2011; 2013) near the Transantarctic Mountains, Antarctica. In summary, Dadic et al.'s collected samples cover a range of ice conditions along two blue ice area (BIA) transects; cores from depths down to 0.94 m were analysed by a combination of microCT analysis, specific surface area (SSA) derived estimates and caliper-and-scale measurements. We have carried out a homogenisation exercise of these and observations from the wider literature to determine sets of internally consistent parameters linked to bulk properties, including the effective scattering coefficient to retain generality. Those referred to further are shown in Table 1; more details are provided in Appendix B.

## 2.2 Photon Initialisation

Individual photons are tracked from their incidence on the air-ice interface ($z = 0$) until they return to the atmosphere, are absorbed, or pass below a depth where their contribution is negligible, here, $z = -16$ m. Each photon is initialised with a Cartesian position and unit direction vector. The $x$ and $y$ coordinates are randomly drawn from a uniform distribution in the range [0, 1] corresponding to a 1 m × 1 m square on the upper interface, with the initial $z$ coordinate being set to zero. To simulate photons from a diffuse sky (or its diffuse component) the direction vector is first expressed as incident azimuth and zenith angles. The azimuth angle, $\phi$, is drawn from a uniform random distribution in the range [0, $2\pi$] whereas for the zenith

angle, $\theta$, the projected area $dA \cos \theta_i$ decreases with increasing angle $\theta_i$. Therefore, so as to maintain the constant radiance definition, the number of photons emitted within a particular angular interval must also decrease. The cumulative probability distribution from which zenith angles are drawn randomly is given by (McNicholas, 1928; Mayer, 2009):

$$P(\theta_i) = 2 \int_0^{\theta_i} \cos \theta_i' \sin \theta_i' \, d\theta_i' = \sin^2 \theta_i. \qquad (3)$$

There is an additional factor of $\sin \theta_i$ due to the three-dimensional geometry. For any direct solar beam component, the direction vector is set appropriately.

Specular reflections at the surface are dealt with by calculating the wavelength and angular dependent reflection coefficient for an unpolarised beam according to Fresnel's equations of reflectance (Hecht, 1987). If a uniform random number in the interval [0, 1] is less than the calculated reflection coefficient then the photon is marked as being returned to the atmosphere.

If it is greater, then the photon is considered to have passed into the ice and the direction vector is updated in line with the angle of refraction.

## 2.3 Photon Interactions

For photons that have passed into the ice the next stage is to calculate the distance before a scattering or absorption event occurs. We construct a cumulative exponential distribution where the mean free path represents the total extinction of the

photon from its straight line path $(k_{sca} + k_{ext})^{-1}$ and sample randomly from this on the interval [0, 1]. Then the single scattering albedo $\omega_0 = k_{sca}/(k_{abs} + k_{sca})$ determines whether the photon is scattered or absorbed: with a probability $\omega_0$ scattering occurs, whilst absorption occurs with a probability $1 - \omega_0$ (Mayer, 2009). This is equivalent to the physical process but used here as it is computationally faster. To demonstrate this equivalence we constructed two cumulative exponential distributions with mean free paths $k_{abs}^{-1}$ and $k_{sca}^{-1}$, and sampled randomly from these in the interval [0, 1].

Whichever produced the smallest result, equivalent to the shortest distance travelled, was the event that is considered to have occurred. Once the event type and the path is determined, the photon position is updated from the direction of travel and distance.

The updated position is checked at each iteration against the local boundaries. First the $x$ and $y$ coordinates are constrained to stay within the limits set at the surface by applying the periodic boundary condition: $x_{i+1} = x_i \, mod \, 1$, and likewise for $y$.

Furthermore, if the updated photon position exceeds a depth of 16 m, it is flagged as such and no longer tracked. If the updated position is found to lie above the ice-air interface, a random number on the interval [0, 1] is compared to the Fresnel reflection coefficient to determine whether it will be transmitted or reflected. If transmission to the atmosphere occurs, the photon packet is marked as such and no longer tracked. If it is reflected back down by the surface, the $z$ coordinate of its position and direction are negated and tracking continues.

If the event corresponds to absorption this is flagged and the photon is no longer tracked. If the interaction is a scattering event then a new direction of travel is calculated, defined by a deflection angle and an azimuthal angle w.r.t. the original direction of travel. The deflection angle is calculated using a Henyey-Greenstein phase function (Henyey and Greenstein,

1941) with the asymmetry parameter $g$, calculated from Mie theory (Bohren and Huffman, 1983) (Fig. 2). The combination of bubble radii and wavelengths included in our analysis corresponds to a size parameter range of between 440 and 5800. Following Pulli et al. (2013) the deflection angle, $\theta$, is then calculated from

$$\cos\theta = \frac{1}{2g}\left[1 + g^2 - \left(\frac{1-g^2}{1-g+2g\chi}\right)^2\right],\tag{4}$$

where $\chi$ is a uniformly distributed random number in the interval [0, 1]. The rotationally symmetric azimuthal angle, $\phi$, is calculated as

$$\phi = 2\pi\chi.\tag{5}$$

From these two angles (which define the scattering event) and the original direction vector, an updated direction cosine vector is calculated (Wang et al., 1995). The process described in this sub-section is repeated until each photon is returned to

the atmosphere, passes below the lower boundary, or is absorbed. Whilst for conceptual reasons the algorithm has been described above as following a single photon, in the model we make use of MATLAB's vectorisation capabilities and typically track $10^4$ photons at a single wavelength, using array indexing only to advance the positions of those that are still in flight.

**2.4 Photon Counting**

As well as recording the final positions of the photons we track them throughout their multiply scattered passage through the ice. In order to do so we record the depth of each photon and whether the direction of its flight is positive (downwards) or negative (upwards) at every step. This allows us to determine the downward and upward irradiance fluxes respectively as fractions of the total number of photons that were initially released. No cosine weighting of the angle to the surface normal is required as it is implicitly included in the photon energy (Mayer et al., 2010; Jacques, 2011). In addition the $z$ coordinates of

absorbed photons can be used to calculate the fractional absorption as a function of depth (here only used as an internal consistency check) and the total fraction of photons returned to the atmosphere (the albedo). We repeat the process at wavelength intervals of 10 nm from 280 nm to 2800 nm. For results integrated over the complete solar shortwave band, the single-wavelength results are interpolated to a 1 nm grid, and then weighted by the incident spectral irradiance, before summing. Note also that whilst the model internally uses a $z$ coordinate that becomes more negative with distance into the

ice, we treat the depth, $d$, as positive parameter from section 3 onwards, increasing below the surface.

**2.5 Model Validation and Limitations**

To validate the described model and test its predictive skill we follow the example of Light et al. (2003) who relied on the four-stream results of Grenfell (1991). For the outputs of interest to us there are three key scenarios in common. The first is a conservative non-refractive domain where we calculate the albedo and transmissivity of a horizontally infinite slab at a range

of optical depths, $\tau$, [1, 2, 5, 10, 20, 50] and asymmetry parameters, $g$, [0.00, 0.50, 0.75, 0.90, 0.95], taking $k_{sca}$ to be 25 m$^-$ $^1$, and set the physical thickness of the slab, $l$, appropriately from $\tau = k_{sca}l$. Next, a conservative refractive case where we

add a lower boundary to our model and set the refractive index of the slab to be 1.31. Finally, a refractive non-conservative case, where we use Light et al. (2003)'s values of $k_{abs}$ for wavelengths of 500 nm to 1000 nm, set $g = 0.95$, $k_{sca} = 250 \text{ m}^{-1}$ and retain a value for the refractive index of 1.31.

Doing so we find the discrepancy between the albedo and transmissivities calculated with the method described herein and the four-stream solution presented in Light et al. are typically ~0.5 % in all three cases. The albedo results for the conservative non-refractive and refractive cases are shown in Fig. 3a and Fig. 3b respectively.

Internally we also check the reproducibility of repeated runs to ensure a stable solution has been reached. For an example wavelength of 600 nm the choice of tracking $10^4$ photons produces $E_\downarrow$ profiles that are consistent at the level of 0.75% (the standard deviation of 5 repeated runs, averaged over depths from 0 m to 2 m). When calculating broadband parameters a total of $2.53 \times 10^6$ photons are tracked, and accordingly this value is found to reduce to 0.041% over the same range of depths.

Whilst the Monte Carlo model produces reproducible results and shows good agreement across the range of optical parameters used by Light et al. (that cover the range of absorption and asymmetry parameters exhibited by blue ice areas) there are some limitations in regards real-world applications. First of these is that all bubbles are assumed to be spherical and thus their single scattering behaviour is governed by Mie theory. This is a useful simplification but in reality individual samples and individual bubbles within them will deviate from perfect spheres. This will affect the asymmetry parameter, and consequently the observed attenuation, but to answer the broader question we choose not to further specify the bubble geometry beyond the assumption that it is spherical. Neither do we consider any vertical or spatial inhomogeneity of bubble density that clearly exists — on a small spatial scale blue ice bubbles form at higher densities along cracks although these cracks do not typically show any preferential orientation. To maintain the general applicability of the results we therefore consider a continuous medium with spherical bubbles, using typical values that are relatable to bulk ice parameters. We note that vertical variations in the asymmetry parameter, the effective bubble radius, and their density could be dealt with by relatively small adaptations to the code.

The model also assumes that the ice-surface is planar. In reality the surfaces of blue ice fields are often scalloped or roughened which would complicate the modelled environment. We anticipate that such surface roughening would generally lead to a reduction in the incident downwelling radiation on most facets, but the effect on the upwelling irradiance is more difficult to assess — where the length-scale of the ice surface geometry is large in comparison to the mean scattering length of photons, locally the surface would appear flat, in line with our planar assumption. A more detailed analysis of this point is therefore left for a future study. We also do not account for any partial covering by a wind-blown snow layer. On the whole blue ice layers are largely free of snow, but we estimate that an intermittent snow layer of depth 5 cm, covering approximately 10 % of the surface would reduce the visible irradiance incident on the upper ice surface by 5 % to 10 % on average (Perovich, 2007). We also anticipate some enhancement of the incident irradiance under cloudy skies due to multiple reflections between clouds and the ground. This mechanism may have not been fully captured by the use of a

broadband albedo input to the atmospheric radiative transfer calculation; it is expected to be of a similar magnitude to that caused by the absence of a snow layer, but in an opposing direction.

## 3 Monte Carlo model results

Our modelling assumes a prescribed description of the incoming solar irradiance spectrum, and estimates of the bubble
number concentration and effective radius. In this study we use a solar irradiance based upon the Frontier Mountain Blue Ice Area, Antarctica [72.95° S, 160.48° E]; full details of its calculation are provided in the Appendix A. Suitable independent estimates for Antarctic Blue Ice Area bubble concentrations and effective radii are stated in Table 1, and background information regarding their calculation is provided in Appendix B.

### 3.1 Spectral considerations *vs.* depth and irradiance enhancement

Assuming fully diffuse sky conditions (that is the incident irradiance has no separate direct beam component), we now apply our Monte Carlo model to four air bubble parameter sets as detailed in Table 1.

In all cases the spectral attenuation by ice is controlled by the interaction of the spectrally-independent scattering coefficient, $k_{sca}$, and the absorption coefficient in pure ice, $k_{abs}$. The latter is strongly wavelength dependent (for reference see later in Fig. 8), varying from between $10^3$ m$^{-1}$ and $10^5$ m$^{-1}$ for $\lambda > 1430$ nm, to a minimum of $6.4 \times 10^{-4}$ m$^{-1}$ at $\lambda = 390$ nm.

Consequently we observe the well-known rapid absorption of photons at the longest wavelengths first, and a shift of the peak in the attenuated spectrum towards shorter wavelengths (Fig. 4). For the no cracks parameter set, corresponding to the least scattering ($k_{sca} = 102.2$ m$^{-1}$) and the largest mean free path of 0.01 m, the solar signal is reduced below 1 mW m$^{-2}$ nm$^{-1}$ for all solar $\lambda > 1351$ nm by a vertical depth of 0.01 m.

Notably for UV and the shortest visible wavelengths there is an enhancement of the subsurface downwelling irradiance, $E_\downarrow$,
so that it is greater than the incident irradiance, $E_0$. Intuitively this is unexpected as the downwelling irradiance is greater than that incident on the ice surface, but it has been previously noted in the literature (Jiang et al., 2005). The enhancement is a result of the change in refractive index at the air-ice interface, combined with the higher refractive index medium exhibiting both scattering and relatively low absorption. Photons that enter the ice are scattered by air bubbles or other contaminants (first order scattering contribution), and assuming a low absorption coefficient, eventually return to strike the
air-ice interface. At this point photons arrive at the interface from the high refractive index medium, and those with an incidence angle greater than the critical angle will undergo total internal reflection, contributing a further time to the downwelling irradiance (second order). Providing it is not absorbed, a photon will continue to be scattered within the ice and may be reflected repeatedly by the inner surface, contributing multiply to the downwelling irradiance (third and higher orders). Likewise the upwelling irradiance will be correspondingly enhanced, and thus the net solar flux across the ice-air
interface does not change and energy is conserved (as noted in Jiang et al., 2005). The energy available for absorption by small contaminants, inclusions, or heating of the ice itself is, however, increased by this enhancement process.

To provide an upper limit for the enhancement we assume a semi-infinite refractive ice slab that scatters but does not absorb photons. For photons incident on the upper surface of the ice-air interface a fraction, $T$, will be transmitted into the body of the ice and contribute to the downwelling irradiance. In the absence of absorption all these will return to impact on the inner side of the interface. A fraction, $R$, of these will then be reflected downward to contribute a second time. Extending this

argument to multiple reflections we find a maximum enhancement factor of:

$$\sum_{n=0}^{\infty} TR^n = \frac{T}{(1-R)} \,. \tag{6}$$

To evaluate this expression, we assume diffuse incident radiation fields on both the upper and lower surfaces of the interface and apply Fresnel reflection and transmission coefficients; for a solar-spectrum weighted refractive index of ice of 1.306, this gives $T = 0.938$ and $R = 0.450$. Consequently the downwelling irradiance can be increased by a factor of up to 1.706

times the incident irradiance at the ice-air interface. For all UV and visible wavelengths the potential enhancement lies between 1.706 and 1.796. For longer wavelengths the absorption by pure ice increases and multiple scattering, and with it the enhancement process, is greatly reduced.

Our Monte Carlo results fall within these theoretical limits (e.g. Fig. 4), with a maximum enhancement factor of 1.735 at $\lambda = $ 390 nm compared to the idealised result of 1.743. At 550 nm the enhancement factor is reduced to 1.573 due to the

increasing absorption. In comparison Jiang et al. (2005) noted an enhancement factor of 1.32 at 550 nm, but this was for a slab 1.7 m thick underlain by sea water and with a lower mean porosity of $< 1 \%$. Both transmission losses through the lower boundary and reduced single scattering albedo ($k_{sca}/(k_{abs} + k_{sca})$) reduce the enhancement.

### 3.2 Effect of bubble parameters

The same interaction between the scattering coefficient and the absorption coefficient that controls the wavelength-

dependence of irradiance also results in a strong wavelength dependence in the total albedo. There are two contributions to the overall albedo of blue ice: the direct, specular reflection according to Fresnel and the contribution from internally back-scattered photons that return to and escape from the surface. The first only exhibits a weak dependence on wavelength, but the latter relies on photons entering the ice not being absorbed before they travel sufficiently far to be scattered back to the surface. Consequently for $\lambda > 1440$ nm the albedo has no internally scattered component, but for shorter $\lambda < 400$ nm the low

absorption coefficient results in an albedo that rises to within ~2 % of unity (Fig. 5). Though scattering by bubbles is a necessary part of the process, we note that there is a relatively small spectral region that is sensitive to such changes: at 820 nm a factor of four increase in $k_{sca}$ alters the spectral albedo from 0.29 to 0.52, an increase of 79 %. At the shortest ($< \sim 500$ nm) and longest wavelengths ($> \sim 1200$ nm) the spectral albedo dependence is much less. As a result the solar irradiance weighted (or broadband) albedo is less dependent on the scattering coefficient, varying from 0.516 for the no cracks

parameter set, to 0.621 for the lower parameter set based on Bintanja's (1999) 850 kg m$^{-3}$ density end point; the solar-irradiance weighted albedo for the mean dataset is 0.605. The range of these results (0.516 to 0.621, 0.585 to 0.621 excluding the no cracks parameter set) includes the albedo estimate used as an input to the libRadtran calculations, and

notably the albedo for our mean dataset agrees well with this initial estimate. The range of values also compares favourably with field measurements, but, for our three representative parameter sets, is narrower: Bintanja (1999) quotes an observed range of 0.56 to 0.69, whilst Dadic et al. (2013) cites an overall range of 0.55 to 0.65 from several earlier BIA studies. In contrast Dadic et al.'s own observations from three specific BIA locations (mean of clear sky and cloudy albedos) show a shift to higher values (0.61 to 0.67).

From a visual perspective it is notable that the spectral variation of the albedo implies that the characteristic colour of bubbled ice is predicted (see highlighted region in Fig. 5): high albedos are found at wavelengths associated with the human eye's short-wavelength (blue) cone response, reducing at visible wavelengths associated with the green cone response, and then still further at wavelengths where the long-wavelength cone is preferentially sensitive (red). In addition, this reinforces the point that blue ice environments with smaller scattering coefficients, $k_{sca}$, (due to fewer or smaller bubbles) result in the most saturated colours, and consequently, show increased fluxes below the surface. This relationship can be inverted for use in the field: the perceived saturation of blue ice can be used as a visual proxy for the density and porosity of blue ice, the amount of scattering, and also potentially used as a rough assessment of the penetration of solar shortwave radiation and inclusions (see later in Sect. 6) into the ice.

The wavelength-integrated results which are normalised by the downwelling irradiance incident on the surface, $E_0$, (which here has been calculated to be 302.5 W m$^{-2}$) are shown in Fig. 6. Again we observe the expected quasi-exponential fall off with depth, as when considering the process from a spectral perspective. Likewise the solar integrated Monte Carlo results also predict the presence of a sub-surface enhancement: for the downwelling flux this is seen down to depths of ~3 cm, with maximum enhancements up to 1.31 times the incident irradiance. A corresponding feature is also observable in the fractional mean irradiance, $(E_\downarrow + E_\uparrow)/2E_0$ (Fig. 6d). Further, the higher scattering, lower density parameter sets show an increased enhancement close to the surface coupled with increased attenuation, causing higher irradiance fluxes close to the surface and reduced fluxes at depth. Additionally we observe a sharp reduction from the peak downwelling irradiance immediately below the surface which is attributed to the rapid absorption of longer IR wavelengths. We also note that the upwelling flux is a substantial fraction of the downwelling flux, almost reaching parity below depths of 1 m.

### 3.3 Dependence on SZA and geographic location

The results presented so far have relied on the assumption that the incoming solar flux arrives from a diffuse hemispherical sky (with no direct component). Atmospheric and other inputs were selected according to a specific location, the Frontier Mountain Range, Antarctica. To test the sensitivity of the presented results to differing assumptions we have rerun the Monte Carlo model to include partitioning between direct and diffuse solar components and at a range of solar zenith angles (SZA). The model has also been implemented with surface elevations, cloud optical depths and solar zenith angles appropriate for a range of geographic locations across both polar regions. In short we find that once partitioning between diffuse and direct components is properly accounted for, the attenuation of the downwelling and upwelling fluxes shows a weaker dependence on SZA. The relationship is most clearly expressed in the spectral albedo at IR wavelengths, while broadband albedos range

between 0.597 and 0.618 for the most representative solar zenith angles of 49° to 69°. Likewise there is only a moderate dependence when specific geographic inputs are chosen — the most obvious effect is a reduction in the IR part of the incident solar flux for locations which exhibit large cloud optical depths or airmass factors. Accordingly, higher latitude, lower elevation and cloudier locations experience somewhat reduced attenuation of the downwelling and upwelling fluxes
per unit of incident flux. This is a secondary effect outweighed by the presence of a cloud layer that will reduce the total irradiance incident on the ice surface.

Further details can be found in the supplementary material.

## 4 Inclusions

So far the Monte Carlo model has been used to investigate the impact of varying bubble radii, number concentration and
solar zenith angle on the propagation of solar irradiance into blue ice; we now apply it to investigate the energy that impacts upon and is absorbed by inclusions within the ice. Accordingly, we adjust the model to count photons whose path intersects with a defined volume element that represents the inclusion. For computational reasons we restrict this test to photons whose positions fall within a set distance of the centre of the inclusion. The path between one scattering event and the next is then subdivided at a granularity of <1mm depending on target size and the photon is treated as absorbed if any point along this
path lies within the inclusion volume. The absorbed photon is added to the downwelling or upwelling count as appropriate. To model a range of possible inclusions we construct spherical, planar, and ellipsoidal geometries. Following Evatt et al. (2016), we choose dimensions appropriate for englacial meteorites augmented by one smaller and one larger geometry (see Appendix C for further details). A single inclusion is defined during each model run to ensure independence: we calculate the fractional downwelling and upwelling irradiance incident on the inclusion at 10 geometrically spaced depths.
Figure 7a shows the fractional irradiance incident on the set of spherical inclusions whilst Fig. 7b shows the same for planar inclusions. The most prominent feature is that, for both cases, the fractional irradiance per unit surface area is substantially lower than without an inclusion. We attribute this to self-shadowing of the diffuse radiation field: downwelling photons once absorbed by the inclusion cannot be scattered up, and then down to contribute multiply to the irradiance. Despite this, the fractional irradiance absorbed by the inclusions is still markedly greater than the singly scattered contribution (also shown in
grey on Fig. 7a and 7b) as the inclusion acts as a sink for photons in its vicinity. This self-shadowing can also be seen to be dependent on both the geometry and dimension of the inclusion: when a target has larger extent in the $x$-$y$ plane compared with the scattering mean free path, it becomes increasingly unlikely that photons from one side of the inclusion will impact upon the opposing surface. Consequently, the mean irradiance incident upon the target is reduced. For a large inclusion close to the surface the downwelling irradiance is similar to the transmitted (singly scattered) fraction of the incident solar signal
as few additional upwelling photons can enter the region between the air-ice interface and the upper surface of the inclusion. For smaller inclusions near the surface the downwelling irradiance increases above the singly scattered contribution, but to a lesser degree than at depth.

For absorbing inclusions, the energy balance may lead to the surrounding ice reaching melting point. Once it does any inclusion denser than water will move downwards under gravity and be capped by a water layer above. When the porosity of ice is > 2.9 % (Battino et al., 1984), the air previously trapped in bubbles will form a layer above the meltwater and create a pair of interfaces: one ice-air and one air-meltwater. These two interfaces reflect a fraction of the diffuse downwelling

irradiance that would have otherwise reached the inclusion. Making use of the spectral ice and water refractive indices (Hale and Querry, 1973) and applying Fresnel reflection coefficients, we find that 55.0 % of the diffuse downwelling irradiance field will be transmitted from the ice into the air. At the air-water interface 93.4 % of this is further transmitted into the meltwater where it will continue unimpeded to the inclusion. In total the presence of an air layer could reduce the downwelling irradiance incident upon the inclusion to 51.4 % of the expected value, though its horizontal extent and

geometry would also have to be taken into account.

The processes of total internal reflection by an ice-air boundary within the ice and self-shadowing also relate to the measurability of the predicted irradiance fluxes. Specifically, attempts to measure the irradiance by insertion of an optical detector into the ice would have to account for both these points. The degree of self-shadowing would be a function of both the size and geometrical shape of the detector, whilst the transmission across the interfaces between the ice, a (partial) air

layer, and the outer envelope of the detector would need to assessed carefully. If they were not included the irradiance fluxes within the body of the ice would be underestimated.

## 5 Comparison to analytic solutions and curve fits

Whilst the described Monte Carlo model aims to replicate the radiative transfer processes occurring within the ice accurately, it is computationally intensive. For each combination of bubble parameter sets and SZA, we follow and track the paths of

approximately $2.53 \times 10^6$ photons, which requires ~15 hours of runtime on a modern desktop PC. Accordingly, there is a utility to being able to replicate these results by analytical means.

Following Marchesini et al. (1989) we can define an effective attenuation coefficient, $K_{eff}$, for a medium where both anisotropic single scattering and absorption are in operation:

$$K_{eff} = \sqrt{\left(3k_{abs}^2 + 3k_{abs}k_{sca}(1-g)\right)}, \tag{7}$$

where the other symbols have the same meanings as previously. In the low scattering limit, this reduces to:

$$K_{eff} = \sqrt{3}k_{abs}, \tag{8}$$

whilst the $\sqrt{3}$ multiplier can be understood as the inverse of the direction cosine, $\bar{\mu}_z$, which has been averaged over each hemisphere as appropriate for an isotropic radiation field. In Fig. 8 we plot the spectral variation of $K_{eff}$ alongside the e-folding distances derived from the Monte Carlo results for the Dadic BIAs no cracks parameter set and a diffuse solar

irradiance. However the near surface downwelling irradiance field is not isotropic as assumed by Eq. (8), but primarily lies within a vertically-orientated cone defined by surface refraction, which implies a larger mean value of $\bar{\mu}_z$, In line with this

we note better agreement between the values of $K_{eff}$ from Monte Carlo e-folding distances and calculated via Eq. 8 at $\lambda >$ 1200 nm, where absorption dominates scattering, if the first multiplier in Eq. (7) is reduced to ~2 to account for this more directional field (Fig. 8). For single wavelengths this construct gives good agreement with the Monte Carlo results, save for the shortest wavelengths and the largest absorption coefficients where discrepancies occur due to the vertical grid interval

and overall size of the model domain.

In light of this, it is tempting to consider the form of the spectrally integrated attenuation curves in Fig. 6c as quasi-exponential, thus imitating the form of the well-known Lambert-Beer exponential relation for attenuation through an absorbing medium. The Lambert-Beer exponential relation holds at a single wavelength: longer wavelengths exhibiting high attenuation, and shorter wavelengths having lower attenuation. Using an integrated form of the Beer Lambert decay function

is common practice (Cuffey and Paterson, 2010; Evatt et al., 2016), with attenuation values typically around 2.5 m⁻¹, as this leads to mathematically tractable estimates for ice temperatures and melt rates (Evatt et al., 2016). However, this approach is somewhat crude and does not explicitly account for the absorption differences in wavelength, e.g. as shown in Fig. 8. To help overcome this, Bintanja et al. (1997) used an exponential-based model which assumed that all of the longer wavelength energy was absorbed at the surface, and that only the shorter wavelengths were able to penetrate to greater depths. Yet this

approach automatically fails to provide explicit information as to the irradiance in the uppermost few centimetres of the ice. One might be tempted to overcome this through use of a double exponential, with the two exponential coefficients being representative for short and long wavebands. However, the absorption coefficients do not fall into two clear ranges and we therefore expect a double exponential fit to underestimate the solar flux at intermediate depths, and conversely to over-estimate the flux at smaller and greater depths. As such, we suggest a triple exponential function to approximate the solar

energy flux attenuation. We now describe how such a function can be parameterised.

As seen in Sect. 3, the attenuation of the irradiance depends upon bubble size and distribution. If the attenuation for distinct ice samples was highly different, then before an analytical approximation could be found, one would first have to solve the full presented Monte Carlo model. Fortunately, plots of the irradiance against depth for the results of Fig. 6c, when scaled against their own surface albedo, show very similar attenuation profiles (Fig. 9) — clearly the result holds less well for the

no cracks data set. As such, one need not necessarily run the Monte Carlo code for each new ice sample. Instead, assuming the ice in question is reasonably generic, one can use a triple exponential function that has been best-fitted to the scaled data sets presented here. For example, if the Dadic BIAs no cracks data is omitted, then the mean scaled net downwelling irradiance, $M_f = \|E_\downarrow - E_\uparrow\|$, has the curve fit against depth in metres, $d$:

$$M_f(d) = 0.084e^{-1.93d} + 0.431e^{-16.9d} + 0.485e^{-195d}. \tag{9}$$

Here all of the coefficients sum to unity so that, if the albedo, $a$, and incoming solar irradiance, $E_0$, are known, then the net irradiance, $N_f = E_\downarrow - E_\uparrow$, at a depth, $d$, can be approximated as:

$$N_f(d) = (1 - a) \cdot E_0 \cdot M_f(d). \tag{10}$$

In so doing, the unique information regarding bubble size and distribution is still present within this equation, for it manifests itself in the size of the albedo, $a$ (where surface broadband albedo estimates are commonly collected field measurements). It is interesting to note that the slowest attenuation coefficient within $M_f$ is, at 1.93 m$^{-1}$, close but lower than the value of 2.5 m$^{-1}$ commonly used by Bintanja et al. (1997; 1999; 2000) and somewhat smaller in magnitude than the 3.3 m$^{-1}$ assumed by Liston et al. (1999) in their constant bulk extinction coefficient simplification. Furthermore by considering the net irradiance as a triple exponential fit rather than a single exponential suggests that there is an increased flux at the shallowest and greatest depths whilst at moderate depth there is a reduction compared with what has been previously assumed.

## 6 Application to Antarctic Meteorites

As noted in the introduction, this study was conceived in order to provide a better understanding of the glacial subsurface radiative field in regards the vertical movement of meteorites through Antarctic blue ice. This is reflected in our choice of inclusion dimensions. In Evatt et al. (2016) the attenuation of solar radiation through ice and the absorption of solar radiation by a meteorite were modelled using the Lambert-Beer law. However the 1D treatment of the problem and simplifications in the assumed radiation field used in that study warrant a more involved investigation. The Monte Carlo model and results described in Sect. 2 to 5 are a core part of that investigation. However computational limitations mean that it is not practical to include the dynamical behaviour of meteorites directly into the Monte Carlo model. Instead we take the two-step approach described below, considering the 3D temperature distribution around static meteorites, and then applying this to a dynamic 1D model.

When considering the hypothesized sinking of englacially transported meteorites, the key parameter is the temperature of the lower (basal) interface between the meteorite and ice. If this rises to 0 ℃, then sinking occurs. The first step, in order to estimate the basal temperature and its distribution around the meteorite, is to construct a static 3D finite difference scheme — a one-dimensional heat equation being inadequate as the meteorite's presence introduces a horizontal inhomogeneity. This scheme considers the complete energy balance of the system and, in addition to the solar shortwave flux, includes longwave radiation, sensible and latent heat fluxes. The triple exponential expression in Eqn. (9) is used to calculate the vertical absorption profile of solar shortwave radiation, whilst the results from Sect. 4 and presented in Fig 7 are used to derive the shortwave radiation incident on the meteorite. Boundary conditions are set following Evatt et al. (2016). Material properties have been refined from those used in Evatt et al. (2016), and are detailed in Mallinson (2019). Surface conditions are selected to be appropriate for the Frontier Mountain region of Antarctica, with 3-hourly values of broadband shortwave and longwave surface fluxes being taken from the SYN1deg CERES product suite (https://ceres.larc.nasa.gov) and 12-hourly air temperature and wind data taken from the ECMWF ERA-Interim reanalysis (Dee et al., 2011). A grid size of 3 mm is required to resolve meteorites sufficiently; in turn the time step is limited to a maximum of 5 s by numerical stability constraints. By applying these conditions and inputs to the 3D finite difference model we produce a time series of the meteorite's basal temperatures for a range of specified depths.

In Fig. 10(a) both the resulting maximum temperatures experienced at the upper and lower surfaces of the meteorite are shown for iron meteorites, and H and L chondritic meteorite classes. This shows that at a fixed depth, meteorites with the lowest iron abundance (L chondrites) produce the lowest basal temperatures; at the upper surface those with the lowest iron abundance produce the highest temperatures. Figure 10(b) gives an alternative view of this result displayed as a cross-section

of the difference in temperature fields surrounding an iron meteorite *vs* an L chondrite. Simply put, when fixed in position, iron meteorites experience the highest basal temperatures. However, this still leaves the question of dynamics unresolved. The second modelling step, to address the dynamical part of the problem, is to develop a numerical one-dimensional implementation of the full heat equation (not a quasi-steady approximation). Its one-dimensional nature gives a clear computational advantage, thus allowing the upward motion of blue ice, the transport and sinking of a meteorite, and a

temporary melt layer to also be included whilst the model is run over several seasons. Initially though the meteorite's depth is fixed whilst the radiation incident upon the upper and lower surfaces of the meteorite is scaled so that the resultant basal temperature of the meteorite matches that predicted from the 3D finite difference model. Finally, the 1D model is run in a dynamical mode with this depth-dependent scaling of the radiation field, permitting the meteorite to produce a melt water layer and sink accordingly within a rising column of blue ice. A more in depth description of these models and inputs can be

found in Mallinson (2019).

Following this approach, and as noted in Evatt et al (2016), we find that seasonal melting is controlled by a meteorite's thermal conductivity, with iron meteorites transferring heat to their lower surface more rapidly than their stony (chondritic) counter-parts. As before, the characteristic sawtooth behaviour is still seen. Whilst ablation is often reduced during the winter, it is still present. Accordingly, the meteorite rises with respect to the ice surface during the winter, and during the

summer when solar heating is active, a meteorite can melt the ice below it and sink, falling faster than the ice surface is ablated. However, our modelling now suggests the process is more nuanced. We find that melt and sinking is initiated slightly later in the year for iron meteorites, but, as their higher thermal conductivity permits basal melting at increased depths, melt also continues later each season. In short, iron meteorites still experience preferential melt and sinking, and hence generally they are predicted to lie below the surface of the ice whilst chondrites remain exposed (see Fig. 11(a)).

However, using the more sophisticated description of radiative transfer described here, coupled to the 3D finite difference model, we predict a minimum depth < 10 cm for cm-scale iron meteorites, significantly less than the ~40 cm calculated in Evatt et al (2016). The potential retrieval of these subsurface iron meteorites is thus predicted to be far easier. The exact depth reached is dependent on the day-to-day forcing factors. Notably though we observe that iron meteorites can be exposed on the surface for part of the year close to the vernal equinox, whilst chondritic meteorites may also sink below the

surface temporarily later in the melt season. The exact durations of the submerged *vs* exposed periods is additionally dependent on ice uplift rate and meteorite geometry. We note a size dependence whereby larger iron meteorites sink deeper, spend less time exposed on the surface, and additionally exhibit more vertical separation from their stony equivalents. This preferential melt result better accords with the relative paucity — but not total absence — of iron meteorites discovered in Antarctica. Specifically, our modelling does not rule out finding some iron meteorites during searches undertaken during

the earlier part of the Austral summer, or given specific meteorite dimensions and surface conditions are met. Likewise, it can explain the observation that sometimes a stony meteorite may be found partially encased in ice (Fig 11(b)), and, importantly, aligns with the observation that the size distribution of iron meteorites is biased to smaller sizes in Antarctica than elsewhere (Harvey, 2003). By extension, this process may explain the restricted altitude range of productive Antarctic

meteorite stranding zones, the number of new discoveries on revisiting a blue ice area, and the absence of meteorite finds from the Greeenland Ice Sheet (Harvey, 2003; Haack et al., 2007)

## 7 Summary

In this study we have undertaken a detailed investigation of shortwave radiative transfer in optically thick ice where

englacial bubbles cause scattering, using a newly developed Monte Carlo model that also includes consideration of inclusions. Our results are primarily applicable to blue ice areas where the surface can be considered horizontal and the ice is relatively compact and snow-free. We have applied an idealised geometry, deriving and using optical properties that are representative of blue ice areas, whilst retaining information about their likely range. This study was motivated by the need for a more extensive investigation into the vertical motion of meteorites within Antarctic blue ice areas. However, the

general conclusions are expected to be relevant for optically thick glacial ice where scattering dominates, noting that in specific cases the internal radiative processes can be complicated by the presence of snow and firn and their microscopic geometries (see also Haussener et al., 2012, who investigated this point for snow), as well as the macroscopic surface geometry. Notwithstanding this simplification, the general results that follow are expected to have wider importance. First we find that there can be an appreciable enhancement of the subsurface downwelling flux, above the level of the

incident irradiance. This was previously observed by Jiang et al. (2005) for a single wavelength of 550 nm, but has been explored here both spectrally and for a solar–integrated quantity. For the normalised units of irradiance that we have used throughout, the integrated solar downwelling flux within the ice can be up to 1.31 times the incident irradiance. There is a corresponding enhancement in the upwelling flux (thus conserving energy). This enhancement is a result of the refractive air-ice boundary overlaying a material volume where scattering dominates, resulting in multiple scattering and internal

reflections. For specific wavelengths where absorption is minimal the enhancement can be as high as 1.735, just less than the theoretical maximum of 1.743 for ice at solar wavelengths in the absence of absorption.

Considering the albedo, our calculations produce the expected wavelength dependence that is interpreted by our visual system as 'blue' (Warren, 2019), with lower porosity ice producing more saturated colours. Thus the perceived saturation of blue ice could be used in the field as a visual proxy for the porosity of ice, its density, and the degree of scattering present,

and moreover, used to coarsely assess the penetration of solar shortwave radiation into the ice. The calculated broadband albedos agree favourably with previous observations, but with a narrower range of 0.585 to 0.621 — though we note that even small changes in broadband albedo can have considerable global consequences over geological timescales (e.g.

Pierrehumbert et al., 2011). Considering the spectral behaviour, the effect is largest at 820 nm where we see a 79 % change when $k_{sca}$ changes by a factor of four. At the shortest and longest wavelengths the spectral albedo shows much less dependence on the scattering coefficient. In addition our model reiterates that there the two components that contribute to the overall albedo: the Fresnel surface reflectance that shows a weak dependence on wavelength, and the strong wavelength

dependence from internally scattered photons.

Our results are predicated on a diffuse incident field under particular assumptions of the surface environment. However, permitting the solar zenith angle to vary, we find a reduced dependence once the incident field is partitioned between diffuse and direct incident components: broadband albedos range between 0.597 and 0.618 for the most representative solar zenith angles (49° to 69°). Considering a range of polar surface environments, the predominant effect is a reduction of the incident

radiative field at IR wavelengths for locations which exhibit large cloud optical depths, or, for clear sky cases, larger airmass factors. As a result, higher latitude, lower elevation and cloudier locations show somewhat reduced attenuation of the normalised downwelling and upwelling fluxes.

For absorbing inclusions embedded within the ice and its internal radiation field, self-shadowing reduces the irradiance incident on the surface of the inclusion. This is a geometric effect, the irradiance reducing for larger inclusions. Conversely

smaller inclusions whose dimensions are less than the mean free path for scattering absorb a greater fraction of the available radiation; for all inclusion sizes downwelling and upwelling fluxes lie between the available irradiance and the single scattering component, the inclusion acting as a local sink for photons. Interpreting the inclusion instead as an optical detector we conclude that both self-shadowing and the introduction of a lower refractive interface between the detector and blue ice must be taken into account when assessing measurements. If they are not taken into account, then our results suggest that

measurements of the irradiance within the ice column would be an underestimate of the actual irradiance present.

Next we assess the Monte Carlo results for ice without an inclusion as a whole and formulate an empirical expression describing the typical behaviour of the net downwelling irradiance in optically thick blue ice. The broadband albedo is separated out, thus leaving the normalised depth dependent aspects to be numerically fitted. The wide range of absorption coefficients exhibited at UV and blue wavelengths, in contrast to those at IR wavelengths, result in a depth dependence that

is inadequately modelled as a single exponential. Instead our result takes the form of a triple exponential function, the two fast decaying components represent the absorption of longer IR wavelengths at shallow depths, whilst the remaining one exhibits a decay constant of 1.93 m$^{-1}$, close to, but lower than, previously noted in the literature. The specificity of the local ice optical properties is retained in, and can be applied to, our expression by setting the broadband albedo appropriately. This new formulation retains the mathematical tractability of previous single exponential approximations (useful for calculating

ice temperatures and melt rates), but more closely adheres to the underlying behaviour, particularly near the surface.

Finally in Sect. 6 the results of the solar shortwave Monte Carlo modelling are applied to a 3D finite difference implementation of the full heat equation to explore the vertical motion of meteorites in Antarctic blue ice areas. Here we find the process is more nuanced than predicted by Evatt et al (2016), with iron meteorites typically residing at shallower depths (< 10 cm *vs* ~ 40 cm), making their potential retrieval much easier. The process is still controlled by the thermal conductivity

of the meteorites, but our results show better agreement with field discoveries — specifically, a limited number of iron meteorite discoveries would be expected, whilst partial submergence of stony (chondritic) meteorites is possible. Notably our modelling shows larger iron meteorites sink deeper, in line with the observed bias of the size distribution of iron meteorites in Antarctica compared to the rest of the world.

## 5  Code availability

Copies of the model routines are available from the corresponding author on request.

## Acknowledgements

The authors would like to acknowledge the follow sources of support that facilitated this study and preparation of the subsequent paper: EPSRC MAPLE Platform Grant No. EP/I01912X/1 (GWE), a summer bursary from the Paneth Meteorite Trust (EH) and Grant No. RPG-2016-349 awarded by The Leverhulme Trust (ARDS, GWE).

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

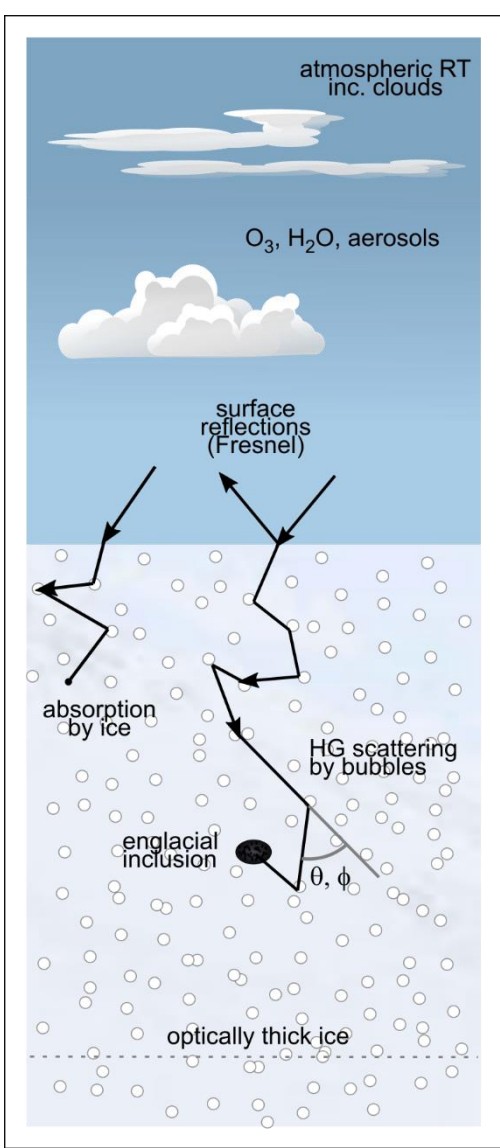

**Figure 1. Schematic illustrating model geometry (not to scale).**

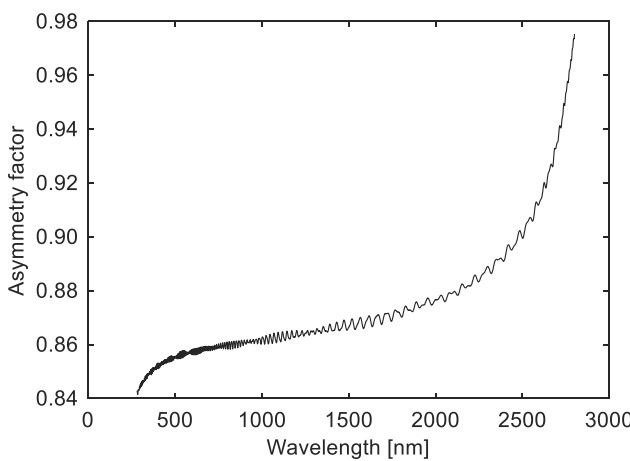

**Figure 2. Spectral variation of asymmetry parameter for spherical air bubbles in ice, calculated from Mie theory (Bohren and Huffman, 1983) for a bubble radius of 198 μm.**

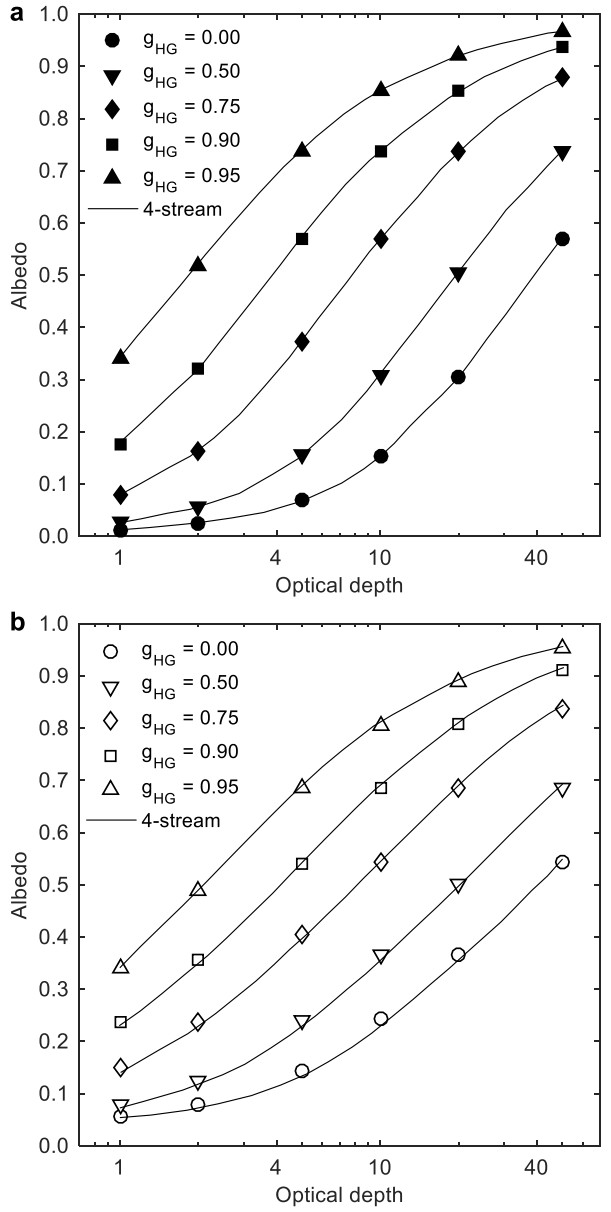

**Figure 3. Comparison of albedo for (a) conservative non-refractive slab and (b) conservative refractive slabs as a function of asymmetry parameter and optical depths. Lines are digitised four-stream results taken from Light et al. (2003) and symbols indicate results calculated from the Monte Carlo code described here. All inputs are chosen to match those in Light et al. (2003).**

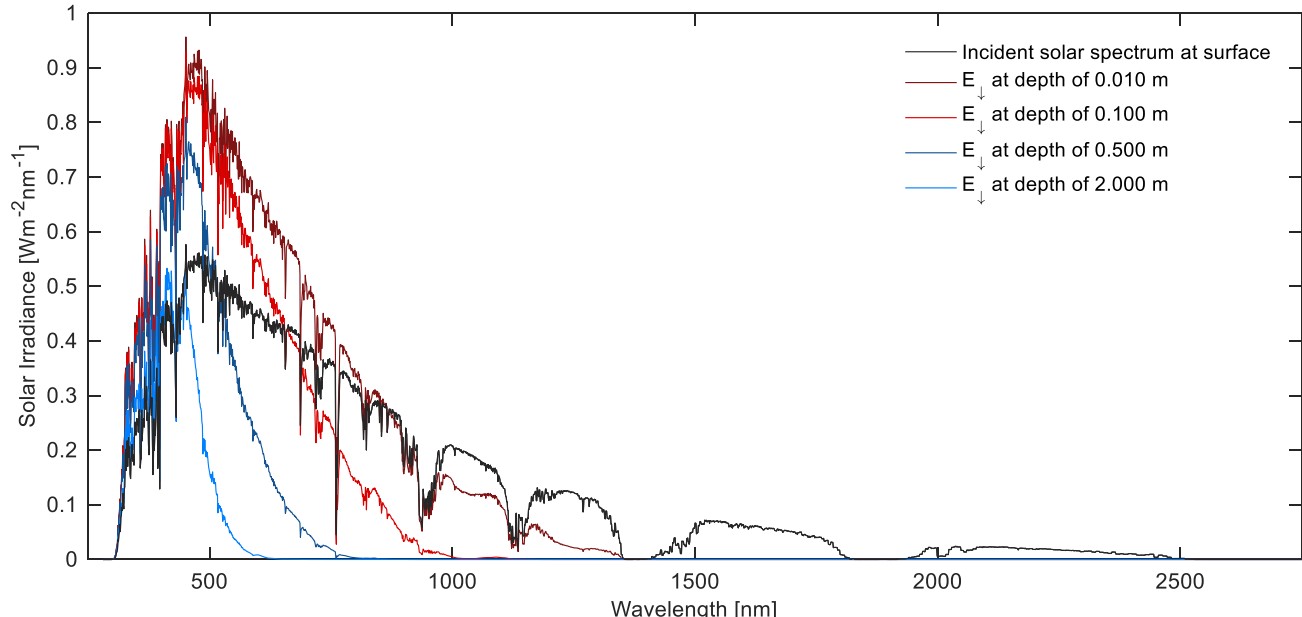

**Figure 4. Spectral downwelling irradiance at the ice surface and at selected depths. The incident solar spectrum is calculated as described in Sect. 2; within the ice, the effective bubble radius is 198 μm and the number concentration is 415 cm⁻³, corresponding to the Dadic BIAs unadjusted no cracks parameter set in Table 1. Subsurface downwelling irradiances enhanced above the incident solar spectrum are seen at visible wavelengths.**

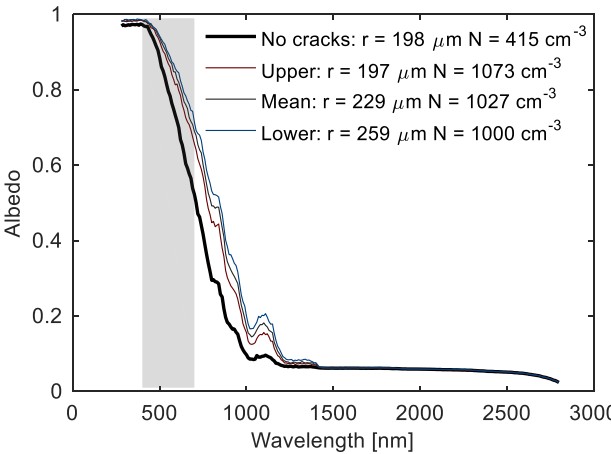

**Figure 5. Spectral albedo calculated for selected bubble parameter sets from Table 1, corresponding to porosities (bubble volume fractions) of 1.35 % for the original Dadic BIAs unadjusted no cracks parameter set ($r_{bub}$ = 198 μm, $N_{bub}$ = 415 cm⁻³); 3.45 % for the upper Dadic BIAs mCT density parameter set ($r_{bub}$ = 197 μm, $N_{bub}$ = 1073 cm⁻³); 5.27 % for the mean Dadic BIAs combined mCT and caliper parameter set ($r_{bub}$ = 229 μm, $N_{bub}$ = 1027 cm⁻³); and 7.31 % for the lower parameter set corresponding to Bintanja's (1999) 850 kg m⁻³ density end point ($r_{bub}$ = 259 μm, $N_{bub}$ = 1000 cm⁻³). Grey shading indicates visible wavelengths.**

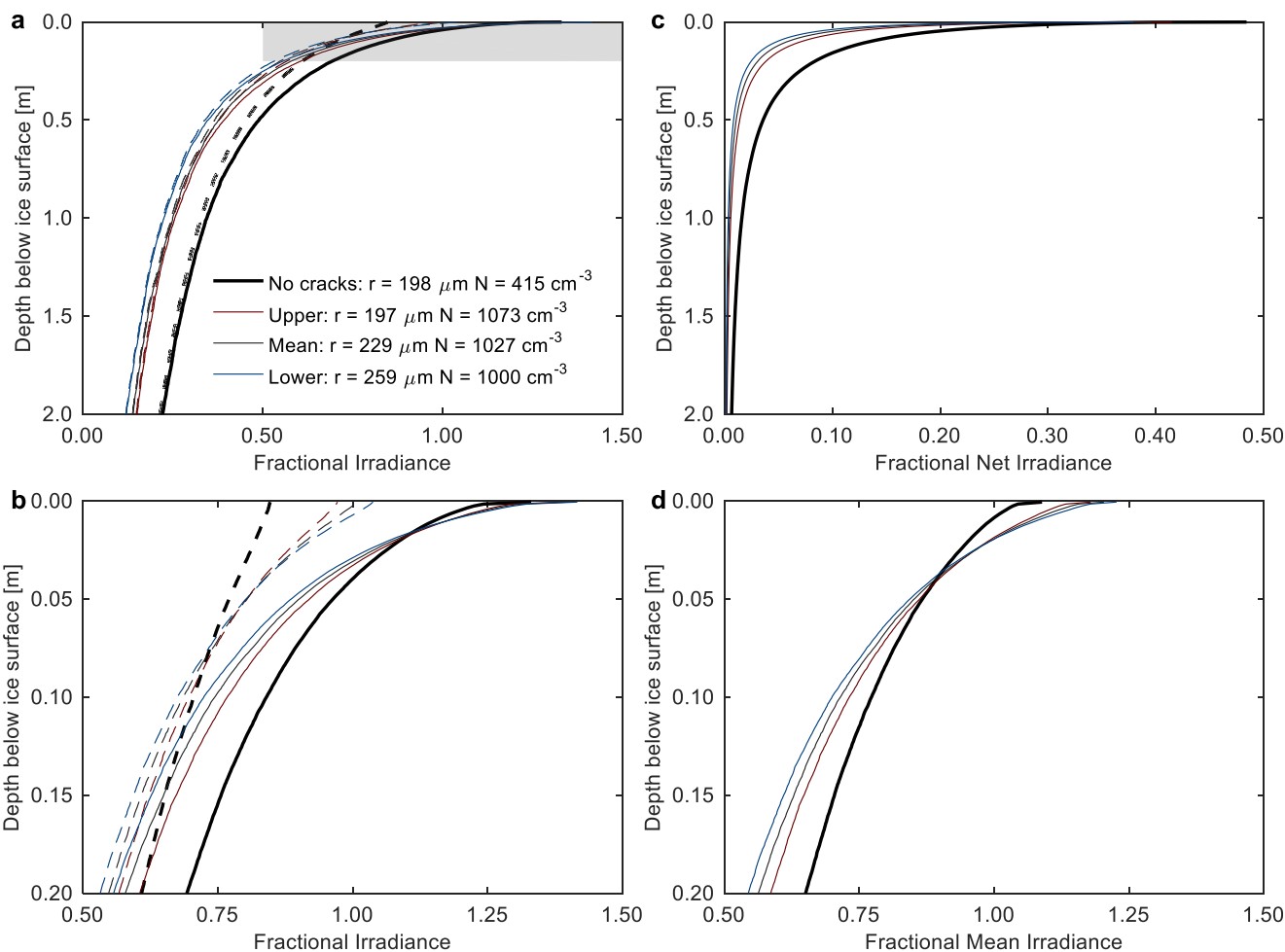

**Figure 6. (a) Fractional solar irradiance variation against ice depth for bubble parameter sets from Table 1. Solid lines show the downwelling irradiance, $E_\downarrow/E_0$, whilst dotted lines show the upwelling irradiance, $E_\uparrow/E_0$. The pairs of lines correspond to porosities (bubble volume fractions) of 1.35 % for the Dadic BIAs unadjusted no cracks parameter set ($r_{bub}$ = 198 μm, $N_{bub}$ = 415 cm$^{-3}$); 3.45 % for the upper Dadic BIAs mCT density parameter set ($r_{bub}$ = 197 μm, $N_{bub}$ = 1073 cm$^{-3}$); 5.27 % for the mean Dadic BIAs combined mCT and caliper parameter set ($r_{bub}$ = 229 μm, $N_{bub}$ = 1027 cm$^{-3}$); and 7.31 % for the lower parameter set corresponding to Bintanja's (1999) 850 kg m$^{-3}$ density end point ($r_{bub}$ = 259 μm, $N_{bub}$ = 1000 cm$^{-3}$). (b) Shows upper grey highlighted area to emphasize subsurface region of Fig. 6a. (c) Shows fractional net irradiance, $(E_\downarrow - E_\uparrow)/E_0$, for same bubble parameter sets, whilst (d) shows fractional mean irradiance, $(E_\downarrow + E_\uparrow)/2E_0$, in the area highlighted in Fig. 6a. In all cases $E_0$ = 302.5 W m$^{-2}$.**

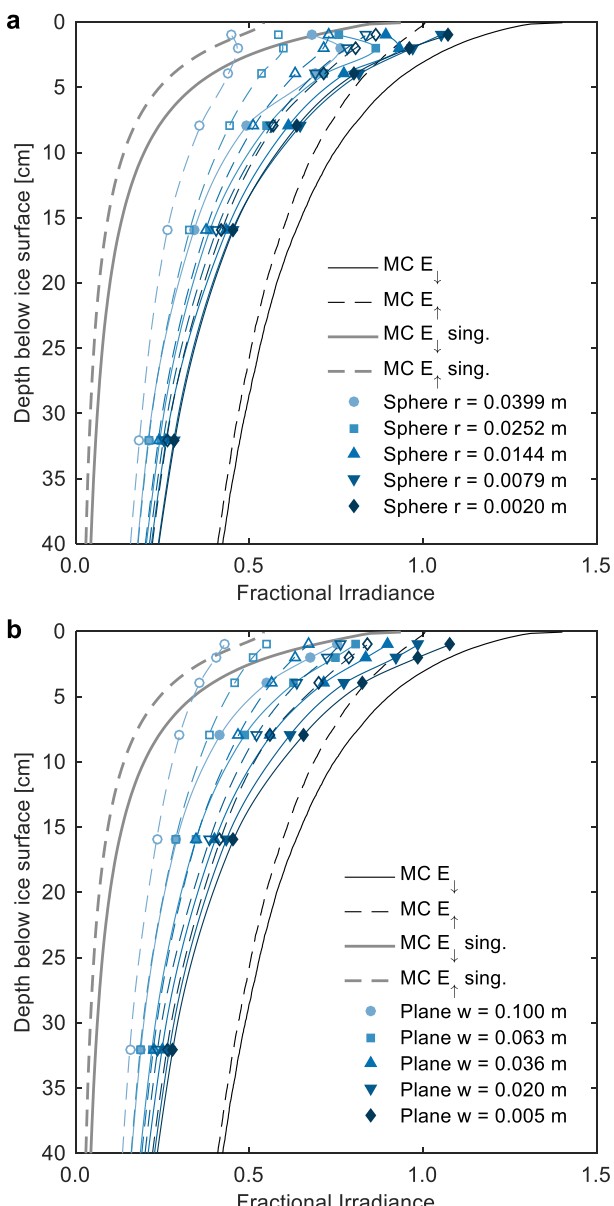

**Figure 7. (a) Fractional solar irradiance absorbed by spherical inclusions against ice depth for a fully diffuse sky and assuming the mean Dadic BIAs combined mCT and caliper bubble parameter set. The solid dark grey line indicates the fractional multiple scattered downwelling components (MC $E_\downarrow$, $E_\uparrow$) within the ice as in Fig. 6, the dotted grey line indicate the upwelling component. Their light grey counterparts show the singly scattered contributions (MC $E_\downarrow$, $E_\uparrow$ sing.). The area-adjusted upwelling and downwelling flux impacting on the spherical inclusions are shown as coloured markers, whilst the solid and dotted lines are interpolations between these. Legend dimensions are radii. (b) Fractional solar irradiance absorbed by planar inclusions against ice depth for a fully diffuse sky and assuming the mean Dadic BIAs combined mCT and caliper bubble parameter set. Markers and lines are as in Fig. 7a. Legend dimensions are the linear extent (width) in the *x-y* plane.**

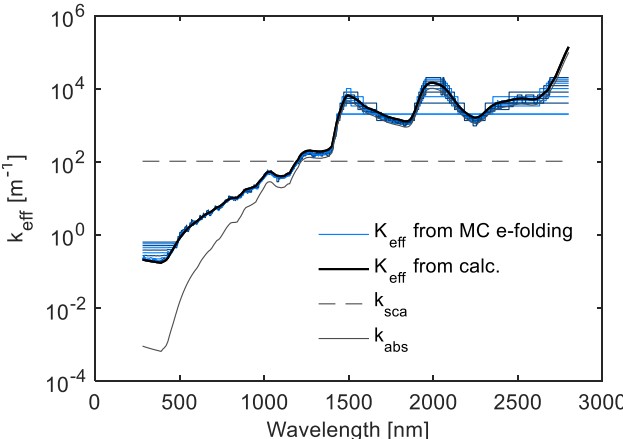

**Figure 8. Spectral variation of the effective attenuation coefficient calculated from e-folding distances (as described in the text) against that calculated from Marchesini et al. (1989). $K_{eff}$ from calculations is shown using a multiplier of 2 for the $k^2_{abs}$ term in Eq. 8. The e-folding distances are calculated as $1/n$ of the depth by which the Monte Carlo downwelling irradiance falls to $e^{-n}$ of its initial values where $n$ are integers in the range [1 … 16]. Also shown is the absorption coefficient for pure ice ($k_{abs}$) and the scattering coefficient ($k_{sca}$) for the Dadic BIAs unadjusted no cracks parameter set ($V_f$ = 1.35 %; $r_{bub}$ = 198 µm, $N_{bub}$ = 415 cm⁻³).**

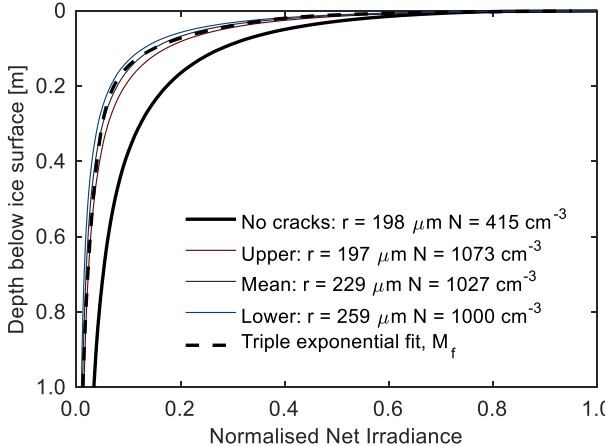

**Figure 9. This figure shows the scaled net irradiance for the four different bubble data sets. The dashed black line, which is almost identical to the grey, shows the triple exponential curve fit, $M_f$.**

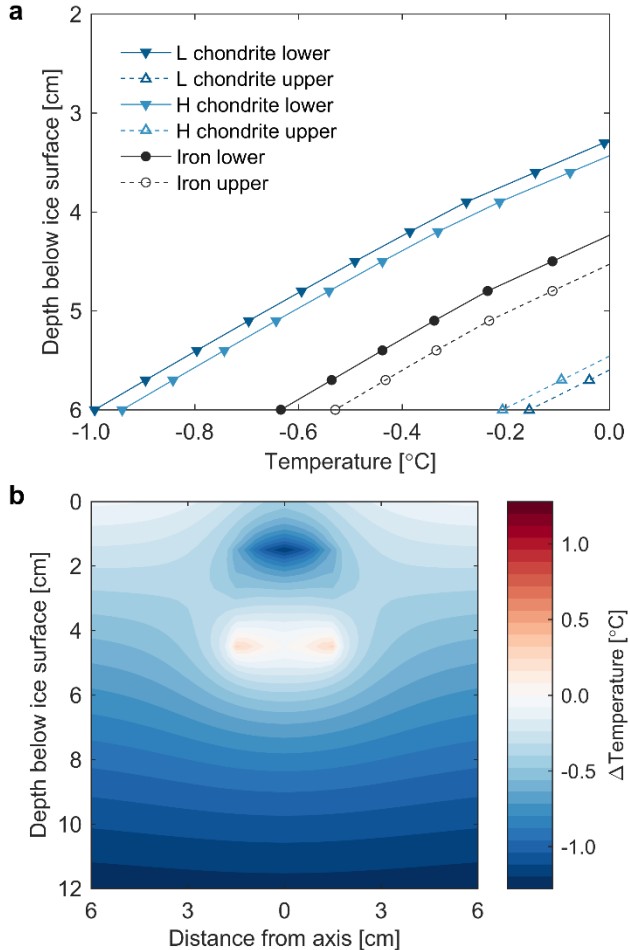

**Figure 10. (a) Maximum temperatures reached at the upper and lower surfaces of L chondrites, H chondrites and iron meteorites when their positions are fixed. Depths are with reference to their centres. In all cases meteorites are 3 cm in height and 3 cm in width. (b) Cross-section of difference in temperature fields surrounding an iron meteorite *vs* an L chondrite when the basal temperature in each case first reaches 0 ºC.**

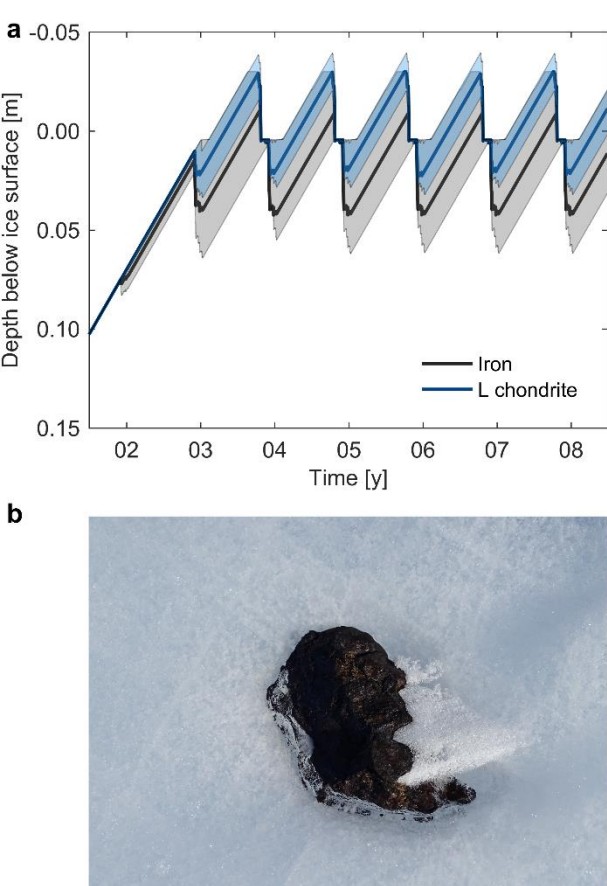

**Figure 11. (a) Time dependence of vertical transport and sinking of meteorites within Antarctic blue ice. Solid lines show depth below surface of meteorite top (height = 3 cm; aspect ratio = 1). Shaded areas (blue for L chondrites and grey for iron) shows range of motion predicted for aspect ratios between 2/3 and 1.5 and heights from 2.1 cm to 4.2 cm. (b) Example of chondritic meteorite partially encased in ice (image credit: Katherine H Joy).**

**Table 1. Summary of bulk ice bubble data referred to in the text and calculated albedos, ranked by increasing bulk density. The Bintanja (1999) (850 kg m$^{-3}$) parameter set is derived from Bintanja's lower estimate of blue ice density and is used as a lower bound parameter set in the text. The mean parameter set is the arithmetic mean of the bubble datasets described in Appendix B and listed in Table B1. The Dadic BIAs mCT density parameters select only samples with microCT density measurements, which are selected to avoid regions with cracks and is used as a parameter set corresponding to a realistic upper bound for density. The Dadic BIAs unadjusted parameter set is also calculated from samples without cracks, but relying on unadjusted microCT density measurements. This last parameter set acts as an outer upper bound with especially low porosities and a density approaching that of pure ice; hereinafter it is referred to the no cracks parameter set.**

| | Density | Porosity, | $r_{bub}$ | $N_{bub}$ | $k_{sca}$ | Albedo |
|---|---|---|---|---|---|---|
| | [kg m$^{-3}$] | $V_f$ [%] | [mm] | [mm$^{-3}$] | [m$^{-1}$] | [0-1] |
| Bintanja (1999) (850 kg m$^{-3}$) [lower] | 850.0 | 7.31 | 0.259 | 1.000 | 422.7 | 0.621 |
| Mean | 868.7 | 5.27 | 0.229 | 1.027 | 340.5 | 0.605 |
| Dadic BIAs mCT density [upper] | 885.4 | 3.45 | 0.197 | 1.073 | 262.4 | 0.585 |
| Dadic BIAs unadjusted [no cracks] | 904.6 | 1.35 | 0.198 | 0.415 | 102.2 | 0.516 |

## Appendices

### A Atmospheric Radiative Transfer

In order to study the passage of solar radiation through bubbled ice, it is necessary to calculate the solar spectrum incident on that surface, considering that many aspects of the transfer are wavelength dependent. Here, the incident spectral irradiance at

the ice surface is calculated using the libRadtran radiative transfer model (Mayer and Kylling, 2005) with relevant atmospheric inputs. These inputs are broadly similar to those in Evatt et al. (2016) used to calculate a climatology of integrated shortwave fluxes, but for completeness we describe them in some more detail here. Internally we use the sdisort radiative transfer solver with pseudo-spherical approximation and the reptran molecular absorption parameterisation, to calculate surface spectral irradiance between 250 nm and 2800 nm at 1 nm intervals. The extra-terrestrial solar spectrum is

from Kurucz (1994). The output altitude is set at 2.04 km with a nominal albedo of 0.62, suitable for the BIA located near Frontier Mountain range in Antarctica [72.95º S, 160.48º E]. The solar zenith angle is set to 66.4º, a representative value that results in a spectrally-integrated irradiance equal to the daily mean on the day of the austral summer solstice.

The clear sky atmosphere profile used is the subarctic summer profile (Anderson et al., 1986) with a climatological total ozone column of 300 DU (Diaz et al., 2004). The spring-summer aerosol profile is taken from Shettle (1989) with the

aerosol optical depth (AOD) at 550 nm scaled to 0.028, the mean of the high and low elevation AOD estimates given by Tomasi et al. (2007). In order to include the effect of clouds, the model is run three times: once with no clouds with the parameters above, once with the addition of a high altitude glaciated cloud, and once with a lower level cloud. The three spectra are then combined linearly in proportion to the relative occurrence estimates of clear skies, high level and lower level clouds to form a single irradiance spectrum. Cloud heights, depths, occurrence frequencies and bulk microphysical properties

are taken from Adhikari et al. (2012).

Care has been taken to be select representative parameter values for the specific BIA locality. However it is anticipated that the resultant spectral shape, though not necessarily the absolute total irradiance, should be also generally applicable to high altitude polar regions (see further in Suppl. S2).

### B Blue ice bubble data

Once the incoming solar radiation reaches the ice surface, the key moderators of radiation through blue ice are the number concentration and radii of bubbles. As noted in the main text there is a paucity of relevant in-situ data and thus we principally rely on a homogenisation of bubble number concentration and radii measurements carried out by Dadic et al. (2013) near the Transantarctic Mountains, Antarctica. Dadic et al.'s samples cover a range of ice conditions along two BIA transects with sample depths down to 0.94 m and are analysed by a combination of microCT analysis, specific surface area

(SSA) derived estimates and caliper-and-scale measurements. As the authors state, the bulk ice densities from the caliper measurements are expected to be underestimates of the density, whilst those from microCT analysis are expected to be over-

estimates. To reconcile these differences and formulate a best estimate of bubble radius and number concentration we first calculate the mean ratio of microCT and caliper densities for samples where both methods have been employed, which allows us to calculate an adjusted density estimate for all samples. In a similar fashion a simple regression is found between measured bubble radii and bulk ice density for the subset of samples undergoing both analysis methods, to give an estimate

for samples lacking radii measurements. In this way we formulate a self-consistent set of densities and radii for the 27 samples available, which together have a mean density of $875\pm12$ kg m$^{-3}$ and a mean bubble radius of $0.236\pm0.028$ mm. A bulk density of pure ice, $\rho_{pure}$, of 917 kg m$^{-3}$ is assumed, and the bulk density, $\rho_{ice}$, effective bubble radius, porosity (or volume fraction of bubbles) $V_{bub}$, and bubble number concentration are related by the following two expressions:

$$\rho_{ice} = \rho_{pure}(1 - V_{bub}) \tag{B1}$$

$$V_{bub} = \frac{4\pi r_{bub}^3 N_{bub}}{3} \tag{B2}$$

To ensure we are not overly reliant on a single set of field campaign data, we apply the density-radii regression and the usual density-porosity-number concentration relations to sub-samples of Dadic et al.'s data and estimates from the wider literature. In this way we aim to represent the range of environments present in BIAs, and are able to calculate an overall mean parameter set and choose example parameter sets corresponding to low and high bulk densities. We additionally select

values corresponding to an outer upper bound for blue ice density and referred to as the 'no cracks' parameter set. Each of these four parameter sets are self-consistent between their estimates of effective bubble radius and number concentration, and, in line with observed bulk densities and porosity values. The parameter sets contributing to the overall mean are listed in Table B1, with additional data being noted in Table B2.

## C Inclusion (Meteorite) Dimensions

We extract dimensions from 94 recently discovered samples detailed in ANSMET newsletters (ANSMET, 2017); all meteorite data are combined to give a mean length $\times$ width $\times$ depth, which we interpret as the lengths of the principal axes $[a\ b\ c]$ of a tri-axial ellipsoid. We find a median value of $a$ to be 2.75 cm with interquartile range of 1.5 cm to 4.8 cm. We also calculate the ratios $b/a$ ($0.775\pm0.015$) and $c/a$ ($0.502\pm0.17$) from which we define a representative lower quartile, a median, and, an upper quartile ellipsoid.

This defines three macroscopic ellipsoidal inclusions with upward facing surface areas of 4.00 cm$^2$, 13.4 cm$^2$, and 40.9 cm$^2$. To ensure the general conclusions are applicable to a wide range of convex absorbing inclusions within the ice, we add one larger target with the same aspect ratios, but an upper surface area of 100 cm$^2$, and one smaller with an upper surface area of 0.25 cm$^2$.

Based on these ellipsoids, five planar and five spherical targets are also defined with linear dimensions and radii chosen so

that their upward facing surface areas match those of the set of ellipsoids.

**Table B1.** Summary of bulk ice bubble data contributing to mean parameter set. Dadic BIAs caliper density parameters selects only samples directly measured by calipers (including samples designated as having cracks). Dadic BIAs combined mCT and caliper parameters incorporate a full set of samples, data being homogenised as described in the text of Appendix B. Dadic BIAs mCT density parameters selects only samples with microCT density measurements (chosen to avoid regions with cracks) selected as representing realistic upper bound for density. Mellor and Swithinbank (1989) parameters are constructed from an estimate that BIAs have a porosity of 6%, combined with the bubble radii-density regression noted in the accompanying text. The Bintanja 1999 (850 kg m$^{-3}$) values are derived from Bintanja's lower estimate of blue ice density and is selected as the lower parameter set. The Bintanja (1999) (865 kg m$^{-3}$) parameter set is derived from the mid-point of Bintanja's range estimate of blue ice density whilst the Bintanja (1999) (880 kg m$^{-3}$) parameter is derived from Bintanja's upper limit for blue ice density. The mean parameter set is the arithmetic mean of these preceding estimates.

|  | Density [kg m$^{-3}$] | Porosity, $V_f$ [%] | $r_{bub}$ [mm] | $N_{bub}$ [mm$^{-3}$] | $k_{sca}$ [m$^{-1}$] |
|---|---|---|---|---|---|
| Dadic BIAs caliper density | 863.3 | 5.86 | 0.236 | 1.064 | 372.3 |
| Dadic BIAs combined mCT and caliper | 875.1 | 4.57 | 0.236 | 0.834 | 290.7 |
| Dadic BIAs mCT density [upper] | 885.4 | 3.45 | 0.197 | 1.073 | 262.4 |
| Mellor and Swithinbank (1989) | 862.0 | 6.00 | 0.238 | 1.059 | 377.7 |
| Bintanja (1999) (850 kg m$^{-3}$) [lower] | 850.0 | 7.31 | 0.259 | 1.000 | 422.7 |
| Bintanja (1999) (865 kg m$^{-3}$) | 865.0 | 5.67 | 0.233 | 1.070 | 365.1 |
| Bintanja (1999) (880 kg m$^{-3}$) | 880.0 | 4.04 | 0.207 | 1.091 | 292.8 |
| Mean | 868.7 | 5.27 | 0.229 | 1.027 | 340.5 |

**Table B2.** Summary of additional bulk ice data bubble data not contributing to mean parameter set in Table B1. Dadic BIAs unadjusted [no cracks] parameter set calculated from samples classed as not including cracks, and relying on unadjusted microCT density measurements. Dadic BIAs mCT density (SSA adjusted) is derived from Dadic BIAs mCT in Table B1, but adjusted for SSA attributed to cracks *vs.* no-crack proportions. Dadic BIAs crack regions selects crack only regions, with a consequently low bulk density.

|  | Density [kg m$^{-3}$] | Porosity, $V_f$ [%] | $r_{bub}$ [mm] | $N_{bub}$ [mm$^{-3}$] | $k_{sca}$ [m$^{-1}$] |
|---|---|---|---|---|---|
| Dadic BIAs unadjusted [no cracks] | 904.6 | 1.35 | 0.198 | 0.415 | 102.2 |
| Dadic BIAs mCT density (SSA adjusted) | 866.8 | 5.48 | 0.197 | 1.703 | 416.5 |
| Dadic BIAs crack regions | 732.0 | 20.17 | 0.237 | 3.618 | 1276.7 |