# Peer review of "Solar radiative transfer in Antarctic blue ice: spectral considerations, subsurface enhancement, inclusions and meteorites"

_The Cryosphere, 2019_

## Referee Comment (RC1) · Stephen Warren (Referee) · 14 Jun 2019

Stephen Warren (Referee)

sgw@uw.edu

Major comment:

The word "inclusions" in the title refers to meteorites hidden just below the ice surface in the Antarctic blue-ice areas, where the searches for meteorites have been carried out for many years. As the authors say in their Introduction, they aim to improve the calculation of "the vertical movement of meteorites through blue ice". The paper would be much more interesting with a little more work, in fact far less work than the examination of different cloud properties described in the Supplement, which was of minor importance. The paper uses Monte Carlo modeling to compute the radiation absorbed

by a meteorite in ice, but then stops just before coming to interesting answers on such questions as (1) How far down below the surface will a meteorite travel in the ice before it stops, as a function of the radius of the meteorite and the incident solar flux? The ice is typically ablating at ∼5 cm/year; can the downward-migrating meteorite outrun the sublimation front? (2) Many meteorites are found on the surface, not melting down into the ice. Is this because they are small enough that the heat from their absorbed radiation is quickly conducted away? These are questions for which the Monte Carlo method is eminently suitable. With just a small additional effort these questions could be addressed. Values of rock density, rock albedo, and thermal conductivity of ice are readily available.

Minor comments:

page 5 line 2. Change Muller to Mullen.

page 6 line 21. "with the dependent variable, the asymmetry parameter g". Henyey-Greenstein is a one-parameter phase function, whose parameter is g; so g is not a "dependent variable". You can just delete the three words "the dependent variable".

page 8 lines 12-13. "enhancement of the incident irradiance under cloudy skies from an albedo feedback mechanism." Change "from an albedo feedback mechanism" to "by multiple reflection between cloud and ground".

page 10 lines 11-12. These three albedo values (51.6, 62.1, 60.5%) should also be included in Table 1.

page 14, Eq. (9). Say that the units of d are meters.

Figure 6a. On the bars at the top, what do the two different colours indicate (blue versus grey)?

Supplement page 1 line 16. "the results for a diffuse sky lies close to the 69° result". I don't see this in Figure S1a; the diffuse sky looks closer to the 49° result.

Figure S1b is unrealistic for the large values of SZA. Even if you ignore diffuse radiation, at SZA=89° the surface roughness present on all blue-ice areas causes the angle of incidence on sunlit surfaces to be considerably less than 89°. I suggest removing the curves for 79 and 89 degrees.

---

## Referee Comment (RC2) · Ruzica Dadic (Referee) · 12 Jul 2019

This paper validates a Monte Carlo method to track photons to calculate reflection and transmission of radiation into Antarctic blue ice. While using the MC method for tracking photons is not exactly new, the authors have made it interesting and relevant by discussing the inclusions of inclusions, such as meteorites. Here I agree with the review of S. Warren, that the study would be much more interesting and novel if the results of the downward movement of those inclusions would be quantified and discussed. As suggested by S. Warren, and considering that many meteorites are found on the surface of the blue ice areas, there either must be some sort of size/material threshold

beyond which the inclusions won't melt into the ice and experience a downward motion, or they are melting much slower than the sublimation in those areas. I would like to see these questions and the questions by S. Warren addressed in a revision, which would make this work more interesting and applicable.

**General comments:**

- Considering that the authors have used the morphological data from Dadic et al. 2013, it would be interesting to see how their results compare to the measured albedos from the same dataset, especially for the validation part. This is particularly relevant because we found that our SSA (bubble size and number density) differs if estimated from Micro-CT (or through the caliper measurements of density) or from the measured albedos and the model.

- P5, L12: I am not convinced that the approach of "homogenization" of our morphological measurements the best solution here. I like that the authors used the "no cracks" data for an upper bound for density, but I think to get a more representative sample, it would have been better to use the lowest density for blue ice for the lower bound, instead of taking means of the data.

- Albedo is usually not given in percent, but as a dimensionless number between 0-1. This should be adjusted, unless there is a particular reason to keep is a percent.

- Generally I found it hard to follow the conclusions from the figures. The figures are referenced, but I often couldn't see where the conclusion is coming from. That could be improved.

[Figure]

**Detailed comments:**

- P2, L23-25: The distinction between snow and firn here is vague and incorrect. Snow starts recrystallizing as soon as it falls, so this distiction is invalid. I suggest either using snow or firn.

- P2, L26: Sea ice also has brine, which has different optical properties than air bubbles.

- P2, L28, This sentence doesn't make sense. Blue ice is formed the same way as glacial ice, which is formed through compression of snow. It's blue because its surface is sublimating and not melting.

- P3, L19-20: This is entirely true. The studies by Mullen and Warren, Light 2003, and Dadic et al. 2013 (studies cited in this paper, and there are probably other studies as well) describe exactly that: spectral albedos and radiative transfer derived from the interaction of solar radiation and embedded bubbles. This sentence should be rephrased: maybe add "by direct ray tracing"?

  Maybe you could also reference Tancrez and Taine 2004 (Direct identification of absorption and scattering coefficients and phase function of a porous medium by a Monte Carlo technique), Haussener et al. 2012 (Determination of the macroscopic optical properties of snow based on exact morphology and direct pore‐level heat transfer modeling) and Farmer and Howell 1998 (Comparison of Monte Carlo Strategies for Radiative Transfer in Participating Media). I'm sure they are others as well.

- P8-9: I can't see the enhancement in Figure 4. Please clarify.

- P8, L3: If you assume that bubble geometry is spherical, why do yo discuss the asymmetry factor? I appreciate that it is acknowledged, but it takes a whole figure

[Figure]

(Figure 2), which has no relevance to this manuscript.

- P10, L9-10 (also in Conclusion and Discussion): I am not sure why a change of albedo from 0.29-0.52 is referred to as "only", and that the albedo is "insensitive" to the scattering coefficient. Small changes in albedo can have a profound effect on the Earth's energy balance. E.g. Pierrehumbert et al. [2011] showed that a change of ice albedo from 0.55 to 0.65 means a factor-of-10 increase in the CO2 mixing ratio required to end the Snowball Earth state. I would like to see this rephrased in the manuscript.

- P10, L18 (also conclusions): It's not exactly a new finding that ice is most re-flective in the blue wavelengths, and I'm not sure why this is one of the conclu-sions. E.g. See Warren et al. 2019 (Green icebergs revisited) and the references therein.

- P14, L15-20: See Haussener et al. 2012 for direct ray tracing in a real geometry.

- P14, L29-31: see comment above about the "insensitivity" of the albedo to scat-tering. Considering that the albedo is between 0-1, it can't be expected that even a fourfold increase in the scattering coefficient would cause a fourfold increase in albedo (which would bring the albedo to above 1). Again, this should be dis-cussed in relative importance of the changes in albedo.

- P15, L1-3: again, not a novel conclusion and relevant papers should be refer-enced.

- P15, L17: Can the main results from Section three be explicitly repeated here, otherwise the reader has to go back through that seciton.

- P15, L17-19: Again, I disagree that there is only a moderate dependence of radiative properties on the zenith angle and bubble parameters. Please have a look at this again.

- Figure 2 can be removed. It doesn't really contribute more than when Mullen and Warren 1998 are referenced.

- Figure 3: I can't see how this Figure is different from Light et al. 2003. I also can't see any difference between figures a) and b). Maybe to make it relevant, you could instead plot the differences between 3a) and 3b) and then discuss the differences.

- Figure 7: Same here, I think to see what the authors are discussing, it would be more helpful to plot the difference between 7a and 7b, otherwise it's hard to draw any conclusions.

[Figure]

---

## Author Comment (AC1) · 20 Aug 2019

We thank both reviewers for the time they have taken to review this manuscript and their useful and constructive comments. It is most appreciated. Below the reviewers' comments are in *italics*, followed by the authors' responses in roman text, and locations of specific changes in the revised manuscript (where not already obvious) follow in [square brackets].

**RC1: Reviewer #1**

*Major comment:*

*The word "inclusions" in the title refers to meteorites hidden just below the ice surface in the Antarctic blue-ice areas, where the searches for meteorites have been carried out for many years. As the authors say in their Introduction, they aim to improve the calculation of "the vertical movement of meteorites through blue ice". The paper would be much more interesting with a little more work, in fact far less work than the examination of different cloud properties described in the Supplement, which was of minor importance. The paper uses Monte Carlo modeling to compute the radiation absorbed by a meteorite in ice, but then stops just before coming to interesting answers on such questions as (1) How far down below the surface will a meteorite travel in the ice before it stops, as a function of the radius of the meteorite and the incident solar flux? The ice is typically ablating at ~5 cm/year; can the downward-migrating meteorite outrun the sublimation front? (2) Many meteorites are found on the surface, not melting down into the ice. Is this because they are small enough that the heat from their absorbed radiation is quickly conducted away? These are questions for which the Monte Carlo method is eminently suitable. With just a small additional effort these questions could be addressed. Values of rock density, rock albedo, and thermal conductivity of ice are readily available.*

We appreciate the reviewer 1's thoughts on this point. This is exactly the core problem we are studying in the wider project, towards which the present study forms an important step. In more detail, our "Lost Meteorites of Antarctica" project is investigating the causes of relative deficiency of iron meteorites in Antarctica, and we have hypothesized whether they might be hidden in a sparse layer below the surface (some 10's of cm below the surface) — a preliminary study was published as Evatt et al., 2016. Following this earlier study a reconnaissance field trip was completed last Austral summer, with the main expedition now in its final planning stages. The hypothesized mechanism is near surface solar heating of the englacial meteorites embedded within upwelling ice flows, and to improve the modelling part of the work we required to better understand the profile of solar absorption by the ice, and the solar radiation available to be absorbed by the meteorites. However the improved modelling work itself is a substantial study — it is not trivial to apply a full 3D free boundary problem — and it could not be done proper justice by inserting an extra section in this current manuscript. It is the subject of a PhD thesis at present and we anticipate a separate manuscript on the meteorite sinking mechanism and its dependencies on their physical properties and dimensions in due course.

--

*Minor comments:*

*page 5 line 2. Change Muller to Mullen.*

Changed [p5, l5].

--

*page 6 line 21. "with the dependent variable, the asymmetry parameter g". Henyey-Greenstein is a one-parameter phase function, whose parameter is g; so g is not a "dependent variable". You can just delete the three words "the dependent variable".*

Changed as suggested. [p6, l24]

--

*page 8 lines 12-13. "enhancement of the incident irradiance under cloudy skies from an albedo feedback mechanism." Change "from an albedo feedback mechanism" to "by multiple reflection between cloud and ground".*

Changed as suggested. [p8, l17]

--

*page 10 lines 11-12. These three albedo values (51.6, 62.1, 60.5%) should also be included in Table 1.*

Added to table 1 as suggested. [Table 1, additional column]

--

*page 14, Eq. (9). Say that the units of d are meters.*

Clarification added.[p14, l8]

--

*Figure 6a. On the bars at the top, what do the two different colours indicate (blue versus grey)?*

We are unsure to what the reviewer is referring at this point. The only 'bar' that appears in figure 6a indicates the highlighted area shown in figure 6b, this is described in the figure caption. In our copy this appears grey, but perhaps it is due a rendering issue.

--

*Supplement page 1 line 16. "the results for a diffuse sky lies close to the 69° result". I don't see this in Figure S1a; the diffuse sky looks closer to the 49° result.*

We thank the reviewer for pointing out this typographical oversight: the diffuse result lies closer to that for a solar zenith angle of 49°. We have amended the text. [Supp. p1, l16]

--

*Figure S1b is unrealistic for the large values of SZA. Even if you ignore diffuse radiation, at SZA=89° the surface roughness present on all blue-ice areas causes the angle of incidence on sunlit surfaces to be considerably less than 89°. I suggest removing the curves for 79 and 89 degrees*

Although our study is idealised in its assumption of a planar surface, even for a roughened / suncupped surface, there will be facets where the incident angle is glancing and approaches 89°. Admittedly these will be small fraction of the total, but the authors feel including the 89° result more clearly demonstrates the impact of differing SZAs, and moreover the need to account for both diffuse and direct components of the incident spectrum (fig S1a vs S1c, fig S1b vs S1d). We have amended the text in light of the reviewer's point. [Supp. p1, l15-16]

**RC2: Reviewer #2**

*This paper validates a Monte Carlo method to track photons to calculate reflection and transmission of radiation into Antarctic blue ice. While using the MC method for tracking photons is not exactly new, the authors have made it interesting and relevant by discussing the inclusions of inclusions, such as meteorites. Here I agree with the review of S. Warren, that the study would be much more interesting and novel if the results of the downward movement of those inclusions would be quantified and discussed. As suggested by S. Warren, and considering that many meteorites are found on the surface of the blue ice areas, there either must be some sort of size/material threshold beyond which the inclusions won't melt into the ice and experience a downward motion, or they are melting much slower than the sublimation in those areas. I would like to see these questions and the questions by S. Warren addressed in a revision, which would make this work more interesting and applicable.*

We refer reviewer 2 to our first response to reviewer 1 on this point. In short, the present study is preparatory work for modelling on this exact problem where we are developing the model of Evatt et al, 2016, and carrying field campaigns to test our hypothesis. Unfortunately we do not feel an adequate description of those methods, nor a full discussion of the results, could be fitted into an additional section in this manuscript.

--

*General comments:*

*• Considering that the authors have used the morphological data from Dadic et al. 2013, it would be interesting to see how their results compare to the measured albedos from the same dataset, especially for the validation part. This is particularly relevant because we found that our SSA (bubble size and number density) differs if estimated from Micro-CT (or through the caliper measurements of density) or from the measured albedos and the model.*

We found that our results lay lower than the albedos measured by Dadic et al. This is noted at p10, ll16-17 in the original manuscript. [p10, ll19-21]

--

*• P5, L12: I am not convinced that the approach of "homogenization" of our morphological measurements the best solution here. I like that the authors used the "no cracks" data for an upper bound for density, but I think to get a more representative sample, it would have been better to use the lowest density for blue ice for the lower bound, instead of taking means of the data.*

Our homogenisation exercise was intended to extract a set of parameters representative of "typical" blue ice, whilst also drawing out parameter estimates for samples with high and low densities. Though the Dadic et al. dataset was exceedingly useful in this regard, we were keen not to rely on a single study location (Allan Hills), to ensure that our results were widely applicable to blue ice areas. For this reason we also incorporated results from the wider literature to construct the upper and lower datasets. The no cracks dataset stands slightly separate from the other three parameter sets in that it less likely that blue ice of that density would exist in a large bulk sample (no cracks existing for several metres), and thus we consider it an outer bound. In contrast the mean, upper and lower parameter sets aim to capture the range of typical blue ice properties.

--

*• Albedo is usually not given in percent, but as a dimensionless number between 0-1. This should be adjusted, unless there is a particular reason to keep is a percent.*

Albedos have been changed to dimensionless numbers. [p5, ll14-21]

--

*• Generally I found it hard to follow the conclusions from the figures. The figures are referenced, but I often couldn't see where the conclusion is coming from. That could be improved.*

We have reviewed the text in light of this suggestion, making clarifications where appropriate.

--

*• P2, L23-25: The distinction between snow and firn here is vague and incorrect. Snow starts recrystallizing as soon as it falls, so this distiction is invalid. I suggest either using snow or firn.*

We take the reviewer's point that there is a graduation between snow and firn and ice, and have rephrased this sentence to clarify this point. [p2, ll23-26]

--

*• P2, L26: Sea ice also has brine, which has different optical properties than air bubbles.*

This point has been added. [p2, ll28-29]

--

• P2, L28, *This sentence doesn't make sense. Blue ice is formed the same way as glacial ice, which is formed through compression of snow. It's blue because its surface is sublimating and not melting.*

This has been corrected. [p2, l29]

--

• P3, L19-20: *This is entirely true. The studies by Mullen and Warren, Light 2003, and Dadic et al. 2013 (studies cited in this paper, and there are probably other studies as well) describe exactly that: spectral albedos and radiative transfer derived from the interaction of solar radiation and embedded bubbles. This sentence should be rephrased: maybe add "by direct ray tracing"?*

We have clarified the text at this point. [p3, ll21-22]

--

*Maybe you could also reference Tancrez and Taine 2004 (Direct identification of absorption and scattering coefficients and phase function of a porous medium by a Monte Carlo technique), Haussener et al. 2012 (Determination of the macroscopic optical properties of snow based on exact morphology and direct poreâA˘ Rlevel heat transfer modeling) and Farmer and Howell 1998 (Comparison of Monte Carlo Strategies for Radiative Transfer in Participating Media). I'm sure they are others as well.*

We thank the reviewer for the suggestions, and have included a reference to Haussener et al. 2012. [p3, l18]

--

• P8-9: *I can't see the enhancement in Figure 4. Please clarify.*

This has been clarified in the caption of figure 4.

--

• P8, L3: *If you assume that bubble geometry is spherical, why do yo discuss the asymmetry factor? I appreciate that it is acknowledged, but it takes a whole figure (Figure 2), which has no relevance to this manuscript.*

As in Mullen and Warren 1988, here we assume spherical bubbles, and from Mie Theory the asymmetry factor is therefore a function of wavelength (shown in fig. 2). The relevance is that the asymmetry parameter controls the scattering and overall path length of a photon within the ice, and hence acts as is a fundamental control on the spectral volume scattering, albedo, and attenuation of solar radiation within the ice. As our study is based upon understanding these radiative processes in blue ice and distilling this into a simple mathematical model, we feel it is useful to show the complex variation exhibited by the asymmetry parameter that might otherwise be overlooked by a reader less versed in radiative theory.

--

• P10, L9-10 (also in Conclusion and Discussion): *I am not sure why a change of albedo from 0.29-0.52 is referred to as "only", and that the albedo is "insensitive" to the scattering coefficient. Small changes in albedo can have a profound effect on the Earth's energy balance. E.g. Pierrehumbert et al. [2011] showed that a change of ice albedo from 0.55 to 0.65 means a factor-of-10 increase in the CO2 mixing ratio required to end the Snowball Earth state. I would like to see this rephrased in the manuscript.*

This has been clarified in the manuscript. The point we were making is that even for the most sensitive wavelengths the spectral albedo changes less than might be expected given the large change in scattering coefficient — and for shorter and longer wavelengths there is less dependence still. Consequently the broadband albedo is relatively insensitive. This is a slightly different issue than

its importance in the Earth's energy balance, on which we wholly agree with the reviewer. [p10, ll10-14]

--

*• P10, L18 (also conclusions): It's not exactly a new finding that ice is most reflective in the blue wavelengths, and I'm not sure why this is one of the conclusions. E.g. See Warren et al. 2019 (Green icebergs revisited) and the references therein.*

While it is not a new result per se, it provides further validation of the model. It could be that the model results showed a high albedo throughout the visual wavelength range, but it did not. We include this paragraph because it is often not appreciated how spectral radiometric and visual responses relate, how the colour of blue ice is manifested by way of the scattering and absorption processes, or that the perceived colour saturation relates to the scattering coefficient — and as such we believe it is useful to highlight these points in the text.

--

*• P14, L15-20: See Haussener et al. 2012 for direct ray tracing in a real geometry.*

We have added a reference to Haussener et al 2012 at the appropriate point in the text. [p14, ll25-26]

--

*• P14, L29-31: see comment above about the "insensitivity" of the albedo to scattering. Considering that the albedo is between 0-1, it can't be expected that even a fourfold increase in the scattering coefficient would cause a fourfold increase in albedo (which would bring the albedo to above 1). Again, this should be discussed in relative importance of the changes in albedo.*

We refer the reviewer to our response to his earlier comment. The relevant sentences in the summary have been rephrased accordingly. [p15, ll4-7]

--

*• P15, L1-3: again, not a novel conclusion and relevant papers should be referenced.*

In a similar vein to our comments on the visual response, it bears repeating that there are two physical processes that give rise to the albedo: photons do not enter the ice and are reflected by the surface and those that enter the ice, and are scattered before being transmitted upwardly back to the atmosphere. In part this is a validation of the model (newly applied to blue ice); in any event the differing spectral contributions of these two processes is often underappreciated.

--

*• P15, L17: Can the main results from Section three be explicitly repeated here, otherwise the reader has to go back through that seciton.*

This sentence has been rephrased to make it clear which part of the results are being referenced. [p15, l23]

--

*• P15, L17-19: Again, I disagree that there is only a moderate dependence of radiative properties on the zenith angle and bubble parameters. Please have a look at this again.*

We refer to the reduced dependence on solar zenith angle when both direct and diffuse components are considered, and the relatively small dependence (given the range of locations explored) on environmental conditions. The dependence on bubble parameters refers to the mean, upper and lower parameter sets which are intended to represent typical blue ice (that is, excluding the 'no cracks' parameter set). This has been clarified at the appropriate point in the manuscript. [p15, ll24-25]

--

• *Figure 2 can be removed. It doesn't really contribute more than when Mullen and Warren 1998 are referenced.*

See note above on figure 2.

--

• *Figure 3: I can't see how this Figure is different from Light et al. 2003. I also can't see any difference between figures a) and b). Maybe to make it relevant, you could instead plot the differences between 3a) and 3b) and then discuss the differences.*

Figure 3a and 3b shows the model validations for a conservative non-refractive slab and a conservative refractive slab respectively. The difference between them can be seen most clearly by examining the lower left hand part of each plot for the $g_{HG}$ = 0.00 curves. The solid curves are the four stream results as presented by Light et al (and should therefore be identical to those shown in their figure). The individual points are our Monte Carlo results — the good agreement demonstrates that our model is working as expected, and hence the reviewer is correct in noting that our figure 3 looks like the equivalent figure in Light et al.

--

• *Figure 7: Same here, I think to see what the authors are discussing, it would be more helpful to plot the difference between 7a and 7b, otherwise it's hard to draw any conclusions.*

Whilst we present results for the two particular geometries of inclusion, our focus in the text is on the how the fractional irradiance varies with depth and size, rather than the differences between spheres and planes. As such our preference is to show each separately which better aligns with our discussion.

---

## Author Response (AR1)

We thank both reviewers for the time they have taken to review this manuscript and their useful and constructive comments. It is most appreciated. In response to the reviewer's suggestion we have included an additional section how the results presented in Sect. 2 to 5 are applied to the dynamics of meteorites in blue ice, and the core results of that extended study. Additionally, the summary has been extensively reworked to clarify our key results and their significance and to address specific recommendations of the reviewers. We have addressed all the reviewer's point-by-point comments at the relevant part of the manuscript; in the marked up version of the manuscript these are indicated in red.

Below the reviewers' comments are in *italics*, followed by the authors' responses in roman text, and locations of specific changes in the revised manuscript follow in [square brackets].

**RC1: Reviewer #1**

*Major comment:*

*The word "inclusions" in the title refers to meteorites hidden just below the ice surface in the Antarctic blue-ice areas, where the searches for meteorites have been carried out for many years. As the authors say in their Introduction, they aim to improve the calculation of "the vertical movement of meteorites through blue ice". The paper would be much more interesting with a little more work, in fact far less work than the examination of different cloud properties described in the Supplement, which was of minor importance. The paper uses Monte Carlo modeling to compute the radiation absorbed by a meteorite in ice, but then stops just before coming to interesting answers on such questions as (1) How far down below the surface will a meteorite travel in the ice before it stops, as a function of the radius of the meteorite and the incident solar flux? The ice is typically ablating at ~5 cm/year; can the downward-migrating meteorite outrun the sublimation front? (2) Many meteorites are found on the surface, not melting down into the ice. Is this because they are small enough that the heat from their absorbed radiation is quickly conducted away? These are questions for which the Monte Carlo method is eminently suitable. With just a small additional effort these questions could be addressed. Values of rock density, rock albedo, and thermal conductivity of ice are readily available.*

We appreciate the reviewer 1's thoughts on this point. This is exactly the core problem we are studying in the wider project, towards which the present study forms an important step. In more detail, our "Lost Meteorites of Antarctica" project is investigating the causes of relative deficiency of iron meteorites in Antarctica, and we have hypothesized whether they might be hidden in a sparse layer below the surface (some 10's of cm below the surface) — a preliminary study was published as Evatt et al., 2016. Following this earlier study a reconnaissance field trip was completed last Austral summer, with the main expedition now in its final planning stages. The hypothesized mechanism is near surface solar heating of the englacial meteorites embedded within upwelling ice flows, and to improve the modelling part of the work we required to better understand the profile of solar absorption by the ice, and the solar radiation available to be absorbed by the meteorites. However the improved meteorite dynamics modelling work itself is a substantial study and requires other environmental considerations to be included besides the solar shortwave and it forms a large part of a recently submitted PhD thesis.

In line with the reviewer's suggestion we have added a section (Sect. 6) to the manuscript on using the results from Sect 2 to 5 to model the dynamics of meteorites near the surface of blue ice areas acting as meteorite stranding zones. We have included a description of our methodology, references to the additional data sources required and used, a discussion of the core results and how they interrelate with prior meteorite collection statistics and observations. Naturally there is additional text also added to the abstract, introduction and summary in line with this new section. In light of the new section we have added the word meteorite to the manuscript title. [p1 l2, p1 l26-28, p4 l8-9, p15 l4 – p16 l34, p18 l23-30]

--

*Minor comments:*

*page 5 line 2. Change Muller to Mullen.*

Changed [p5, l14].

--

*page 6 line 21. "with the dependent variable, the asymmetry parameter g". Henyey-Greenstein is a one-parameter phase function, whose parameter is g; so g is not a "dependent variable". You can just delete the three words "the dependent variable".*

Changed as suggested. [p7, l1]

--

*page 8 lines 12-13. "enhancement of the incident irradiance under cloudy skies from an albedo feedback mechanism." Change "from an albedo feedback mechanism" to "by multiple reflection between cloud and ground".*

Changed as suggested. [p8, l27-28]

--

*page 10 lines 11-12. These three albedo values (51.6, 62.1, 60.5%) should also be included in Table 1.*

Added to table 1 as suggested. [Table 1, additional column]

--

*page 14, Eq. (9). Say that the units of d are meters.*

Clarification added.[p14, l8]. Units added in following paragraph [p14 l32-34]

--

*Figure 6a. On the bars at the top, what do the two different colours indicate (blue versus grey)?*

We are unsure to what the reviewer is referring at this point. The only 'bar' that appears in our copy of figure 6a indicates the highlighted area shown in figure 6b, this is described in the figure caption. In our copy this appears grey, but perhaps it is due a rendering issue.

--

*Supplement page 1 line 16. "the results for a diffuse sky lies close to the 69° result". I don't see this in Figure S1a; the diffuse sky looks closer to the 49° result.*

We thank the reviewer for pointing out this typographical oversight: the diffuse result lies closer to that for a solar zenith angle of 49°. We have amended the text. [Supp. p1, l17]

--

*Figure S1b is unrealistic for the large values of SZA. Even if you ignore diffuse radiation, at SZA=89° the surface roughness present on all blue-ice areas causes the angle of incidence on sunlit surfaces to be considerably less than 89°. I suggest removing the curves for 79 and 89 degrees*

Although our study is idealised in its assumption of a planar surface, even for a roughened / suncupped surface, there will be facets where the incident angle is glancing and approaches 89°. Admittedly these will be small fraction of the total, but the authors feel including the 89° result helps demonstrates the impact of differing SZAs, and moreover the need to account for both diffuse and direct components of the incident spectrum (fig S1a vs S1c, fig S1b vs S1d). We have amended the text in light of the reviewer's point. [Supp. p1, l16-17] We plan further development of the Monte Carlo model that will allow questions regarding the influence of surface texture to be explored in detail.

**RC2: Reviewer #2**

*This paper validates a Monte Carlo method to track photons to calculate reflection and transmission of radiation into Antarctic blue ice. While using the MC method for tracking photons is not exactly new, the authors have made it interesting and relevant by discussing the inclusions of inclusions, such as meteorites. Here I agree with the review of S. Warren, that the study would be much more interesting and novel if the results of the downward movement of those inclusions would be quantified and discussed. As suggested by S. Warren, and considering that many meteorites are found on the surface of the blue ice areas, there either must be some sort of size/material threshold beyond which the inclusions won't melt into the ice and experience a downward motion, or they are melting much slower than the sublimation in those areas. I would like to see these questions and the questions by S. Warren addressed in a revision, which would make this work more interesting and applicable.*

We refer reviewer 2 to our first response to reviewer 1 on this point. In short, the reviewed manuscript is preparatory work for modelling on this exact problem where we develop the model of Evatt et al, 2016, and carrying field campaigns to test our hypothesis. To model the dynamics of meteorites directly in the Monte Carlo model is problematic due to computational restrictions and the necessary time interval to maintain model stability. Instead we apply the results of the earlier sections to implementations of the full heat equation. This methodology is described in an additional new section (Sect. 6) where we also present a discussion of the core results and how they interrelate with prior meteorite collection statistics and observations. There is additional text added to the abstract, introduction and summary in line with this new section and an adjustment to the manuscript title. In short our modelling show the primary selection process is due to the differing material characteristics of stony *vs* iron meteorites, but also that the size and geometry has an impact. In reassessing this process we find that iron meteorites are expected to lie < 10 cm below the surface, much shallower than was previously predicted. [p1 l2, p1 l26-28, p4 l8-9, p15 l4 – p16 l34, p18 l23-30]

--

*General comments:*

*• Considering that the authors have used the morphological data from Dadic et al. 2013, it would be interesting to see how their results compare to the measured albedos from the same dataset, especially for the validation part. This is particularly relevant because we found that our SSA (bubble size and number density) differs if estimated from Micro-CT (or through the caliper measurements of density) or from the measured albedos and the model.*

We found that our results lay lower than the albedos measured by Dadic et al. This is noted at p10, ll16-17 in the original manuscript, but more detail has been added to the text. [p10 ll25 – p11 l2]

--

*• P5, L12: I am not convinced that the approach of "homogenization" of our morphological measurements the best solution here. I like that the authors used the "no cracks" data for an upper bound for density, but I think to get a more representative sample, it would have been better to use the lowest density for blue ice for the lower bound, instead of taking means of the data.*

Our homogenisation exercise was intended to extract a set of parameters representative of "typical" blue ice, whilst also drawing out parameter estimates for samples with higher and lower densities. Though the Dadic et al. dataset was exceedingly useful in this regard, we were keen not to rely on a single study location (Allan Hills), to ensure that our results were widely applicable to blue ice areas. For this reason we also incorporated results from the wider literature to construct the upper and lower datasets. The no cracks dataset stands slightly separate from the other three parameter sets in that it less likely that blue ice of that density would exist in a large bulk sample (no cracks existing for several metres), and thus we consider it an outer bound. In contrast the mean, upper and lower parameter sets aim to exemplify the range of typical blue ice properties.

--

*• Albedo is usually not given in percent, but as a dimensionless number between 0-1. This should be adjusted, unless there is a particular reason to keep is a percent.*

Albedos have been changed to dimensionless numbers here and throughout. [especially at p10 l26 – p11 l2]

--

*• Generally I found it hard to follow the conclusions from the figures. The figures are referenced, but I often couldn't see where the conclusion is coming from. That could be improved.*

We have reviewed the text in light of this suggestion, making clarifications to assist the reader.

--

*• P2, L23-25: The distinction between snow and firn here is vague and incorrect. Snow starts recrystallizing as soon as it falls, so this distiction is invalid. I suggest either using snow or firn.*

We take the reviewer's point that there is a graduation between snow and firn and ice, and have rephrased this sentence to clarify this point. [p2, ll31-32]

--

*• P2, L26: Sea ice also has brine, which has different optical properties than air bubbles.*

This point has been added. [p3, ll1-2]

--

*• P2, L28, This sentence doesn't make sense. Blue ice is formed the same way as glacial ice, which is formed through compression of snow. It's blue because its surface is sublimating and not melting.*

This has been corrected. [p3, l3]

--

*• P3, L19-20: This is entirely true. The studies by Mullen and Warren, Light 2003, and Dadic et al. 2013 (studies cited in this paper, and there are probably other studies as well) describe exactly that: spectral albedos and radiative transfer derived from the interaction of solar radiation and embedded bubbles. This sentence should be rephrased: maybe add "by direct ray tracing"?*

We have clarified the text at this point along the lines the reviewer suggests. [p3, ll28-29]

--

*Maybe you could also reference Tancrez and Taine 2004 (Direct identification of absorption and scattering coefficients and phase function of a porous medium by a Monte Carlo technique), Haussener et al. 2012 (Determination of the macroscopic optical properties of snow based on exact morphology and direct poreâAˇRlevel heat transfer modeling) and Farmer and Howell 1998 (Comparison of Monte Carlo Strategies for Radiative Transfer in Participating Media). I'm sure they are others as well.*

We thank the reviewer for the suggestions, and have included a reference to Haussener et al. 2012. [p3, l25]

--

*• P8-9: I can't see the enhancement in Figure 4. Please clarify.*

This has been clarified in the caption of figure 4 and also in the main text. [p9 l18]

--

*• P8, L3: If you assume that bubble geometry is spherical, why do yo discuss the asymmetry factor? I appreciate that it is acknowledged, but it takes a whole figure (Figure 2), which has no relevance to this manuscript.*

As in Mullen and Warren 1988, here we assume spherical bubbles, and from Mie Theory the asymmetry factor is therefore a function of wavelength (shown in fig. 2). The relevance is that the asymmetry parameter controls the scattering and overall path length of a photon within the ice, and hence acts as is a fundamental control on the spectral volume scattering, albedo, and attenuation of solar radiation within the ice. As our study is based upon understanding these radiative processes in blue ice and distilling this into a simple mathematical model, we feel it is useful to show the complex variation exhibited by the asymmetry parameter (and other aspects) that might otherwise be overlooked by a reader less versed in radiative theory.

--

*• P10, L9-10 (also in Conclusion and Discussion): I am not sure why a change of albedo from 0.29-0.52 is referred to as "only", and that the albedo is "insensitive" to the scattering coefficient. Small changes in albedo can have a profound effect on the Earth's energy balance. E.g. Pierrehumbert et al. [2011] showed that a change of ice albedo from 0.55 to 0.65 means a factor-of-10 increase in the CO2 mixing ratio required to end the Snowball Earth state. I would like to see this rephrased in the manuscript.*

This has been clarified in the manuscript. The point we were making is that even for the most sensitive wavelengths the spectral albedo changes less than might be expected given the large change in scattering coefficient — and for shorter and longer wavelengths there is less dependence still. Consequently the broadband albedo is less sensitive. This is a slightly different issue than its importance in the Earth's energy balance, on which we wholly agree with the reviewer. This part has been rephrased, and also in the summary. [p10 ll22-25, p17 ll23-28]

--

*• P10, L18 (also conclusions): It's not exactly a new finding that ice is most reflective in the blue wavelengths, and I'm not sure why this is one of the conclusions. E.g. See Warren et al. 2019 (Green icebergs revisited) and the references therein.*

While it is not a new result per se, it provides further validation of the model. We include this paragraph because it is often not appreciated how spectral radiometric and visual responses relate, how the colour of blue ice is manifested by way of the scattering and absorption processes, or that the perceived colour saturation relates to the scattering coefficient — and as such we believe it is useful to highlight these points in the text. We have clarified the inter-relation between radiometric and visual responses and its potential usefulness in the field w.r.t. our other results, both here and also in the conclusions. With a view to the specific application of Antarctic meteorite field searches, the link between meteorite depth, penetration of solar shortwave and the visual appearance of the ice may prove particularly useful, giving field teams a simple indicator of how deep remotely detected meteorites may lie in a specific area of blue ice. We have clarified the potential usefulness of this in the text. [p11 ll8-11]

--

*• P14, L15-20: See Haussener et al. 2012 for direct ray tracing in a real geometry.*

We thank the reviewer for this suggestion and have added a reference to Haussener et al 2012 at the appropriate point in the manuscript. [p17, l9]

--

*• P14, L29-31: see comment above about the "insensitivity" of the albedo to scattering. Considering that the albedo is between 0-1, it can't be expected that even a fourfold increase in the scattering coefficient would cause a fourfold increase in albedo (which would bring the albedo to above 1). Again, this should be discussed in relative importance of the changes in albedo.*

We thank the reviewer for highlighting this point. The relevant sentences in the summary have been rephrased accordingly. [p17, ll26-28]

--

• *P15, L1-3: again, not a novel conclusion and relevant papers should be referenced.*

We feel it bears repeating that there are two physical processes that give rise to the albedo: photons do not enter the ice and are reflected by the surface and those that enter the ice, and are scattered before being transmitted upwardly back to the atmosphere. In part this is a validation of the model (newly developed and applied to blue ice); in any event the differing spectral contributions of these two processes is often underappreciated. We have clarified in the text that this is not a new result. [p17 l28]

--

• *P15, L17: Can the main results from Section three be explicitly repeated here, otherwise the reader has to go back through that seciton.*

This sentence has been rephrased to make it clear which part of the results are being referenced. [p18, l13]

--

• *P15, L17-19: Again, I disagree that there is only a moderate dependence of radiative properties on the zenith angle and bubble parameters. Please have a look at this again.*

We refer to the reduced dependence on solar zenith angle when both direct and diffuse components are considered, and the relatively small dependence (given the range of locations explored) on environmental conditions. The dependence on bubble parameters refers to the mean, upper and lower parameter sets which are intended to represent typical blue ice (that is, excluding the 'no cracks' parameter set).

This has now been addressed and rephrased at the appropriate point in the manuscript. [p17, ll23-28, p17 l32]

--

• *Figure 2 can be removed. It doesn't really contribute more than when Mullen and Warren 1998 are referenced.*

See response above on figure 2.

--

• *Figure 3: I can't see how this Figure is different from Light et al. 2003. I also can't see any difference between figures a) and b). Maybe to make it relevant, you could instead plot the differences between 3a) and 3b) and then discuss the differences.*

Figure 3a and 3b shows the model validations for a conservative non-refractive slab and a conservative refractive slab respectively. The difference between them can be seen most clearly by examining the lower left hand part of each plot for the $g_{HG} = 0.00$ curves. The solid curves are the four stream results as presented by Light et al (and should therefore be identical to those shown in their figure). The individual points are our Monte Carlo results — the good agreement demonstrates that our Monte Carlo model is working as expected, and hence the reviewer is correct in noting that our figure 3 looks like the equivalent figure in Light et al.

--

• *Figure 7: Same here, I think to see what the authors are discussing, it would be more helpful to plot the difference between 7a and 7b, otherwise it's hard to draw any conclusions.*

Whilst we present results for the two particular geometries of inclusion, our focus in the text is on the how the fractional irradiance varies with depth and size, rather than the differences between spheres and planes. As such our preference is to show each separately which better aligns with our discussion.

[revised manuscript text omitted]

---

## Author Response (AR2)

We thank the referee for the time she has taken to further review this manuscript. We have addressed the reviewer's point-by-point comments at the relevant part of the manuscript; in the marked up version of the manuscript these are indicated in red.

Below the referee's comments are in *italics*, followed by the authors' responses in roman text, and locations of specific changes in the revised manuscript follow in [square brackets].

**Referee #2**

*I have read this much improved manuscript and recommend it for publication after some minor corrections. The added section 6 makes this paper really interesting and relevant and I appreciate the work the authors have done to make it happen.*

*There are only two things that would be a good addition in the paper:*

*1) The authors assume a planar surface, which probably leads to an exaggeration of the enhancement factor. While I understand that this is planned in future work, it would be helpful if the authors discussed what implication this assumption has on the results.*

We have added some text addressing this point at p8 ll24-29. There is an argument that surface roughness and scalloping reduces the average net incident radiation on the blue ice surface – in simple terms explained by "shadowing" of some facets, but perhaps the effect on the subsurface enhancement is more complex. For blue ice where the length-scale of the ice surface geometry is large in comparison to the mean scattering length of photons, locally the surface would appear planar to scattered photons. And, since the radiation field in the ice is diffuse, it is not clear to the authors that that this would change the subsurface enhancement much, the one exception being at any cusps that are present. [p8 ll24-29]

*2) I don't quite understand the processes behind figure 11b, that make the meteorite "rise" at a different time of year than "sink". Assuming that the solar radiation heats the meteorite up and make it sink, and that ablation (sublimation) will make them "rise" closer to the surface (both processes happening at the same time of year); I don't understand the zig-zag line in the figure, which implies that the sinking and rising happen at a different time of the year. Can you please explain this in the text?*

We have added an explanation at p16 ll18-21 which we hope clarifies the point. Though both processes occur at the same time of year (during the summer), the rate of melt is greater than the rate of ablation, so the meteorite sinks. During the winter, solar heating is, of course, absent, and the vertical transport of the meteorite is dependent solely on the motion and ablation of ice (though this often can be reduced during the winter). [p16 ll18-21]

[revised manuscript text omitted]